# Eater cooperates with Multiplexin to drive the formation of hematopoietic compartments

Gábor Csordás[1]*, Ferdinand Grawe[1,2], Mirka Uhlirova[1]*

[1]Institute for Genetics and Cologne Excellence Cluster on Cellular Stress Responses in Aging-Associated Diseases (CECAD), University of Cologne, Cologne, Germany; [2]Molecular Cell Biology, Institute I for Anatomy, University of Cologne Medical School, Cologne, Germany

**Abstract** Blood development in multicellular organisms relies on specific tissue microenvironments that nurture hematopoietic precursors and promote their self-renewal, proliferation, and differentiation. The mechanisms driving blood cell homing and their interactions with hematopoietic microenvironments remain poorly understood. Here, we use the *Drosophila melanogaster* model to reveal a pivotal role for basement membrane composition in the formation of hematopoietic compartments. We demonstrate that by modulating extracellular matrix components, the fly blood cells known as hemocytes can be relocated to tissue surfaces where they function similarly to their natural hematopoietic environment. We establish that the Collagen XV/XVIII ortholog Multiplexin in the tissue-basement membranes and the phagocytosis receptor Eater on the hemocytes physically interact and are necessary and sufficient to induce immune cell-tissue association. These results highlight the cooperation of Multiplexin and Eater as an integral part of a homing mechanism that specifies and maintains hematopoietic sites in *Drosophila*.

*For correspondence:
cgabor@uni-koeln.de (GC);
mirka.uhlirova@uni-koeln.de (MU)

**Competing interests:** The authors declare that no competing interests exist.

## Introduction

In the animal kingdom, the development and differentiation of immune cells is intimately linked to specific spatial compartments. These sites shelter developing immune precursors from external stimuli while providing signals to orchestrate their self-renewal, proliferation and differentiation. During mammalian embryonic development, the hematopoietic stem cells (HSCs) relocate to the fetal liver where they considerably expand in number, before they populate the spleen and the bone marrow in late embryonic stages (*Gao et al., 2018*). Postnatally, transplanted HSCs primarily seed the hematopoietic niches within the bone marrow by recognizing specific signals on the endothelial cells of the blood vessels, and undergoing trans-endothelial migration (*Birbrair and Frenette, 2016*). In the bone marrow, the HSCs attach to endosteal cells, which together form the endosteal HSC niche capable of renewing the entire blood cell pool throughout life (*Birbrair and Frenette, 2016*). Apart from various non-hematopoietic cells which guard HSC behavior, the extracellular matrix (ECM) has been recognized as an essential component of hematopoietic stem cell niches (*Klamer and Voermans, 2014*). The ECM provides a structured microenvironment to the niche and connects to HSCs through integrin-mediated adhesion (*Gattazzo et al., 2014*; *Khurana et al., 2016*). In turn, this mechanosensitive signal feeds back on the stem cells to regulate their proliferative capacity or differentiation (*Choi and Harley, 2012*; *Lee et al., 2013*). Although mounting evidence points to the requirement for the ECM in the homeostatic maintenance of the HSC population (*Klamer and Voermans, 2014*), it remains unexplored if ECM alters the behavior of immune precursors solely by conveying mechanical signals or whether there are receptor-ligand interactions that depend on the recognition of specific ECM components which trigger activation of discrete signaling pathways.

Furthermore, it remains to be uncovered whether the presence of particular ECM proteins on niche surfaces alone can provoke the adhesion and expansion of the immune cells.

In recent years, the fruit fly *Drosophila melanogaster* emerged as an excellent model to study the dynamics of hematopoiesis (*Banerjee et al., 2019*). Similar to mammals, *Drosophila* immune cells, called hemocytes, are present from early embryonic stages, and reside in specific hematopoietic sites during development (*Martinez-Agosto et al., 2007*). In the larval stages, hemocytes form three hematopoietic tissues: the circulation, the lymph gland and the sessile hematopoietic pockets (*Honti et al., 2014*; *Letourneau et al., 2016*). The circulation comprises mostly macrophage-like cells (plasmatocytes) and crystal cells, which participate in the melanization of encapsulated foreign objects (e.g. parasitic wasp eggs) (*Lanot et al., 2001*). These capsules are largely formed by a third type of hemocytes, the lamellocytes, which are not present under homeostatic conditions, but rapidly differentiate upon immune challenge (*Lanot et al., 2001*). Unlike the freely moving cells in the circulation, the lymph gland is a compact multi-lobe hematopoietic organ on the anterior end of the dorsal vessel, where immune cell precursors differentiate into plasmatocytes and crystal cells (*Jung, 2005*; *Krzemien et al., 2010*). Importantly, the lymph gland-derived hemocytes enter the circulation only during pupariation or upon immune challenge such as parasitic attack (*Krzemień et al., 2007*; *Sorrentino et al., 2002*). The sessile hematopoietic pockets are located segmentally along the length of the larva in lateral and dorsal patches contained within epidermis and muscle tissue (*Makhijani et al., 2011*; *Márkus et al., 2009*). The sessile tissue is primarily composed of plasmatocytes, some of which undergo trans-differentiation into crystal cells (*Leitão and Sucena, 2015*). It has been demonstrated that the formation of sessile hematopoietic pockets is orchestrated by sensory neurons of the peripheral nervous system (PNS) that not only attract hemocytes but also support their survival and proliferation in situ by secreting Activin-β, a ligand of the TGF-β family (*Makhijani et al., 2017*; *Makhijani et al., 2011*). Furthermore, plasmatocytes require the cell-autonomous expression of Eater, a phagocytosis receptor of the Nimrod family (*Kocks et al., 2005*), to maintain their attachment to sessile pockets (*Bretscher et al., 2015*; *Melcarne et al., 2019*). The molecular counterpart of Eater on the body wall remains yet unknown. While these mechanisms anchor the immune cells to the epidermis, they do not isolate them, as there is a continuous exchange between circulating and sessile hemocytes (*Honti et al., 2010*; *Lanot et al., 2001*; *Makhijani et al., 2011*). Moreover, in response to various stress insults sessile hemocytes can be rapidly mobilized and enter circulation (*Márkus et al., 2009*; *Vanha-Aho et al., 2015*), while reestablishing the stereotypical pattern of hematopoietic pockets when the challenge ceases (*Makhijani et al., 2011*), highlighting the dynamic nature of the sessile hematopoietic compartment as well as a requirement for homing cues and adhesive surfaces.

Since the sessile hematopoietic pockets are formed in gaps between the larval epidermis and the body wall muscles (*Makhijani et al., 2011*), the immune cells are in intimate contact with the basement membranes covering these surfaces. In fact, hemocytes can relocate to other tissues with pathologically altered ECM composition and/or structure, such as imaginal discs or salivary glands with damaged basement membranes (*Casas-Tintó et al., 2015*; *Hauling et al., 2014*; *Pastor-Pareja et al., 2008*), tumors (*Cordero et al., 2010*; *Kulshammer and Uhlirova, 2013*; *Pérez et al., 2017*) or fibrotic adipose tissues (*Zang et al., 2015*). As a mechanism to cope with such insults and to participate in developmental tissue remodeling, hemocytes secrete ECM proteins, such as Laminins or Collagen IV (*Bunt et al., 2010*; *Töpfer et al., 2019*), as well as ECM processing enzymes or assembly factors (*Martinek et al., 2008*; *Nelson et al., 1994*). However, only little is known about the physical interaction between hemocytes and ECM under either pathological or homeostatic conditions.

Here, we show that the hemocyte-basement membrane interaction is crucial to the formation of the sessile hematopoietic pockets. We demonstrate that the sessile hemocytes require the interaction of the phagocytosis receptor Eater and the Collagen XV/XVIII ortholog Multiplexin in the epidermal basement membrane to maintain their association with the body wall. Importantly, by manipulating the basement membrane composition hemocytes can be redirected to other tissue surfaces, where they function similarly as they do in the sessile pockets.

## Results

### Increased Atf3 levels induce hemocyte attachment to the fat body

The basic leucine zipper (bZIP) domain protein, Activating transcription factor 3 (Atf3), has been defined as a stress-response gene pivotal to the maintenance of immune and metabolic homeostasis (*Chakrabarti et al., 2014*; *Gilchrist et al., 2006*; *Jadhav and Zhang, 2017*; *Rynes et al., 2012*; *Zmuda et al., 2010*), but has also been implicated in the regulation of cytoskeletal dynamics and vesicular trafficking (*Boespflug et al., 2014*; *Donohoe et al., 2018*; *Yuan et al., 2013*). We have reported previously that the fat body-specific overexpression of Atf3 under the *C7-GAL4* driver (*Koyama and Mirth, 2016*) (hereafter abbreviated as *C7>Atf3^{WT}*) induced a lean phenotype and decreased lipid droplet size (*Rynes et al., 2012*). Intriguingly, a closer examination of *C7>Atf3^{WT}* larvae in which the fat body was visualized with a help of *UAS-GFP* and the hemocyte population with *Hml:DsRed* reporter (hereafter abbreviated as *Hml:DsRed, C7>Atf3^{WT}*) revealed a massive attachment of hemocytes to the adipose tissue surface (*Figure 1B, D and E*), a phenotype not detected in control larvae (*Figure 1A, C and E* and *Figure 1—figure supplement 1A*). The fat body-associated hemocytes (hereafter referred to as FBAHs) present in *Hml:DsRed, C7>Atf3^{WT}* larvae were densely packed and often formed contiguous monolayers over the entire length of the adipose tissue (*Figure 1—figure supplement 1B*). In contrast, the characteristic pattern of sessile hematopoietic pockets observed in controls was disrupted in *Hml:DsRed, C7>Atf3^{WT}* larvae (*Figure 1A and B*), while the amount of circulating hemocytes was not significantly affected (*Figure 1—figure supplement 1C*). Importantly, we found that hemocytes also associated with clones of Atf3 overexpressing adipocytes induced by the heat shock FLPout (hsFLPout) technique (*Figure 1F*), suggesting that the attachment is an intrinsic property of the fat body cells with excessive Atf3 levels rather than a consequence of the systemic activation of blood cells. In support of this notion, neither the uptake of GFP-positive material originating from the fat body was observed in the hemocytes, nor signs of an encapsulation reaction, such as adhesion of lamellocytes or tissue melanization, were present in the case of *C7>Atf3^{WT}* fat bodies (*Figure 1—figure supplement 1D–1F*).

Together these data indicate that the hemocytes either actively relocate from the sessile hematopoietic pockets or are redirected from the circulation to preferentially adhere to the fat body. Importantly, the interaction of hemocytes with Atf3 expressing adipocytes is not a result of anti-tissue immune response.

### Redirected hemocytes proliferate and differentiate on the adipocyte surfaces and engage in immune response

The fact that the hemocytes favored the attachment to the *C7>Atf3^{WT}* fat body at the expense of body wall sessile pockets indicated that the immune cells associated with Atf3 overexpressing adipocytes may form a de novo hematopoietic compartment reminiscent of the sessile tissue. To determine whether FBAHs would display features of the natural sessile hemocyte population, we focused on their morphological characteristics, mitotic activity, differentiation and ability to participate in the immune responses. Intriguingly, FBAHs, like sessile hemocytes (*Lanot et al., 2001*; *Leitão and Sucena, 2015*; *Makhijani et al., 2011*; *Márkus et al., 2009*), projected filopodia of varying lengths as well as lamellipodia (*Figure 2A–B*) and proliferated in situ as demonstrated by the Fly-FUCCI in vivo cell cycle reporter (*Zielke et al., 2014*) and immunostaining for a mitotic marker phospho-histone-H3 (pH3) (*Figure 2C–D*).

Staining against the plasmatocyte marker NimrodC1 (NimC1) (*Kurucz et al., 2007a*) and the crystal cell-specific BcF6:GFP reporter (*Tokusumi et al., 2009*) showed that like in the sessile tissue (*Leitão and Sucena, 2015*), the majority of the FBAHs were plasmatocytes, interspersed by a smaller number of crystal cells (*Figure 2E*). Surprisingly, a portion of the BcF6:GFP-positive crystal cells also stained for NimC1 and showed Hml:DsRed levels similar to plasmatocytes, suggesting that these cells are intermediate hemocytes in the process of plasmatocyte-crystal cell trans-differentiation (*Figure 2E–E'*). Moreover, we observed blackened crystal cells among FBAHs (*Figure 2F*) when combining the *Hml:DsRed, C7>Atf3^{WT}* with the *Bc^1* mutation (*Rizki et al., 1980*). Since the runaway melanization cascade results in cell death (*Lanot et al., 2001*; *Neyen et al., 2015*), we speculate that their differentiation likely occurred on the adipocyte surface. Notably, both BcF6:GFP

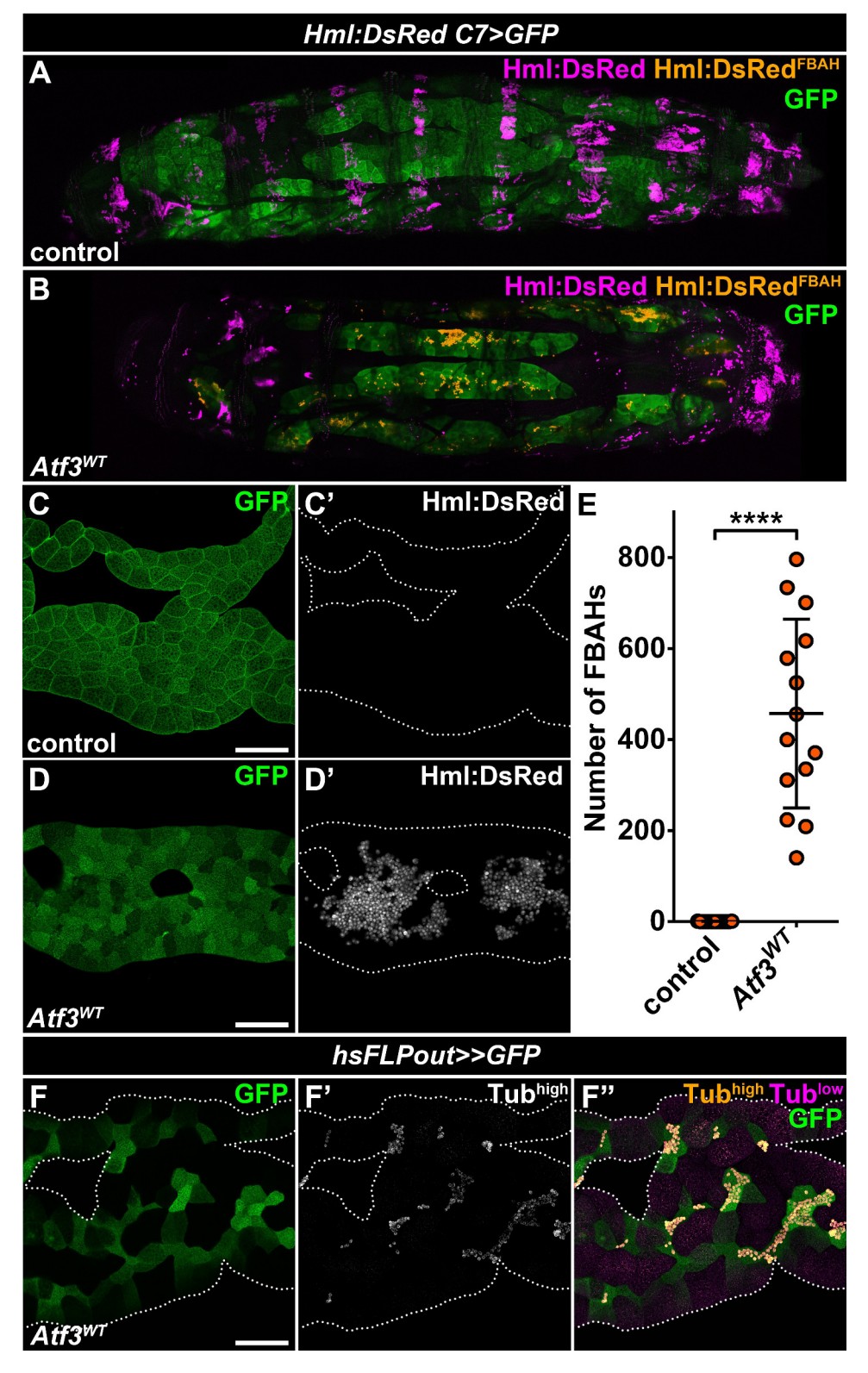

**Figure 1.** Adipose tissue-specific Atf3 overexpression redirects hemocytes to the fat body surface. (A–B) Sessile hemocytes present in control larvae as segmental stripes (A) are redirected to the surface of the fat body overexpressing Atf3 (B). Transgene and GFP expression was driven by the fat body-specific *C7-GAL4* driver, while *Hml:DsRed* marks the hemocytes. The images are stitched from multiple Z-projections, where FBAHs are colored amber, all other hemocytes magenta and the fat body green. (C–D) In control larvae, no hemocytes are present on the fat body (C). Atf3 overexpression

*Figure 1 continued on next page*

*Figure 1 continued*

induces the attachment of hemocytes, which form large clusters on the fat body surface (D). Transgene and GFP expression was driven by the fat body-specific *C7-GAL4* driver, while *Hml:DsRed* marks the hemocytes. (E) Quantification of fat body-attached hemocyte numbers upon Atf3 overexpression with the *C7-GAL4* driver. Data points represent individual fat bodies. Unpaired nonparametric two-tailed Mann-Whitney test was used to calculate p-values. Error bars indicate SD, n = 15, ****p < 0.0001. (F) Clonal overexpression of Atf3 induced with the hsFLPout system causes selective hemocyte attachment (F', white, F'', amber) to the clonal adipocytes (marked with GFP). Hemocytes are identified based on strong Tubulin staining (F'), while the weak Tubulin staining in the fat body was pseudocolored magenta (F''). Fat bodies are outlined with dotted lines (C, D, F). The images are projections of multiple confocal sections. Scale bars: 100 μm (C, D, F). See also *Figure 1—figure supplement 1* and *Figure 1—source data 1*.

The online version of this article includes the following source data and figure supplement(s) for figure 1:

**Source data 1.** Quantification of FBAH number in control and *C7>Atf3^WT^* larvae.
**Source data 2.** Circulating hemocyte counts from control and Atf3 overexpressing larvae.
**Figure supplement 1.** Hemocytes redirected to the fat body are not engaged in anti-tissue response.

expressing and melanized crystal cells in the *Bc^1^* background were mostly located within the FBAH clusters and were surrounded by plasmatocytes (*Figure 2E–F*).

Finally, to evaluate whether the FBAH population can participate in an immune response as previously reported for the sessile hemocytes (*Márkus et al., 2009*; *Vanha-Aho et al., 2015*), control and *Hml:DsRed, C7>Atf3^WT^* larvae were infected with *Leptopilina boulardi* parasitoid wasps. We found that the FBAHs detached from the Atf3 overexpressing adipose tissue 48 hr after parasitic infection (*Figure 2G–I*), and lamellocyte differentiation and capsule formation (*Figure 2J, L, N and P*) was not impaired to a noticeable extent when compared to control larvae (*Figure 2J, K, M and O*). Together, these data support the notion that hemocytes associated with the fat body in *C7>Atf3^WT^* larvae display a hematopoietic program reminiscent of sessile blood cells and engage in the innate immune defense. The *C7>Atf3^WT^* FBAH model thus represents a unique opportunity to dissect cellular and molecular mechanisms of hemocyte-tissue homing.

## Basement membrane accumulation underlies hemocyte association to Atf3 overexpressing fat bodies

The *Drosophila* larval adipocytes are the primary source of the basement membrane components for most internal organs (*Pastor-Pareja and Xu, 2011*), including their own. A crucial step in the formation of basement membranes is the proper deposition and assembly of the ECM components which rely on an intricate cellular and membrane trafficking machinery. Hindering ECM protein secretion and release can lead to fibrotic accumulation (*Shahab et al., 2015*; *Zang et al., 2015*). Given the described role of Atf3 in cellular trafficking in epithelial cells (*Donohoe et al., 2018*), we asked whether the attachment of hemocytes to the Atf3 overexpressing fat body could be caused by changes in the structure or composition of the basement membrane. Indeed, the visualization of the Vkg::GFP reporter, a fusion protein of Collagen IVα2 and GFP (*Morin et al., 2001*), as well as immunostaining against Laminin and the Collagen XV/XVIII-type protein Multiplexin (Mp) revealed accumulation of ECM below the basement membrane in Atf3 overexpressing clonal adipocytes (*Figure 3A–C*). Transmission-electron microscopy (TEM) further showed that in contrast to the mostly smooth cell membranes of control adipocytes (*Figure 3E*), the *C7>Atf3^WT^* fat bodies formed deep pericellular folds containing electron-dense material (*Figure 3F*). These data suggest that the ECM proteins are trapped in pericellular spaces, which might attract hemocytes to the fat body surface.

To establish if there is a causal link between the accumulation of the basement membrane components and the attachment of hemocytes, we decided to interfere with the proper deposition of ECM proteins by silencing the matricellular chaperone SPARC (*Shahab et al., 2015*), and to reduce the amount of specific core ECM components in the *C7>Atf3^WT^* adipose tissue. As expected, inhibition of SPARC enhanced accumulation of ECM aggregates in *C7>Atf3^WT^* fat body cells which was associated with membrane blebbing (*Figure 3—figure supplement 1A–1B*). In contrast, the knockdown of Col4a1 (Collagen IVα1) and Trol (Perlecan) strongly suppressed fat body association of hemocytes, while silencing Vkg partially inhibited blood cell attachment (*Figure 3—figure*

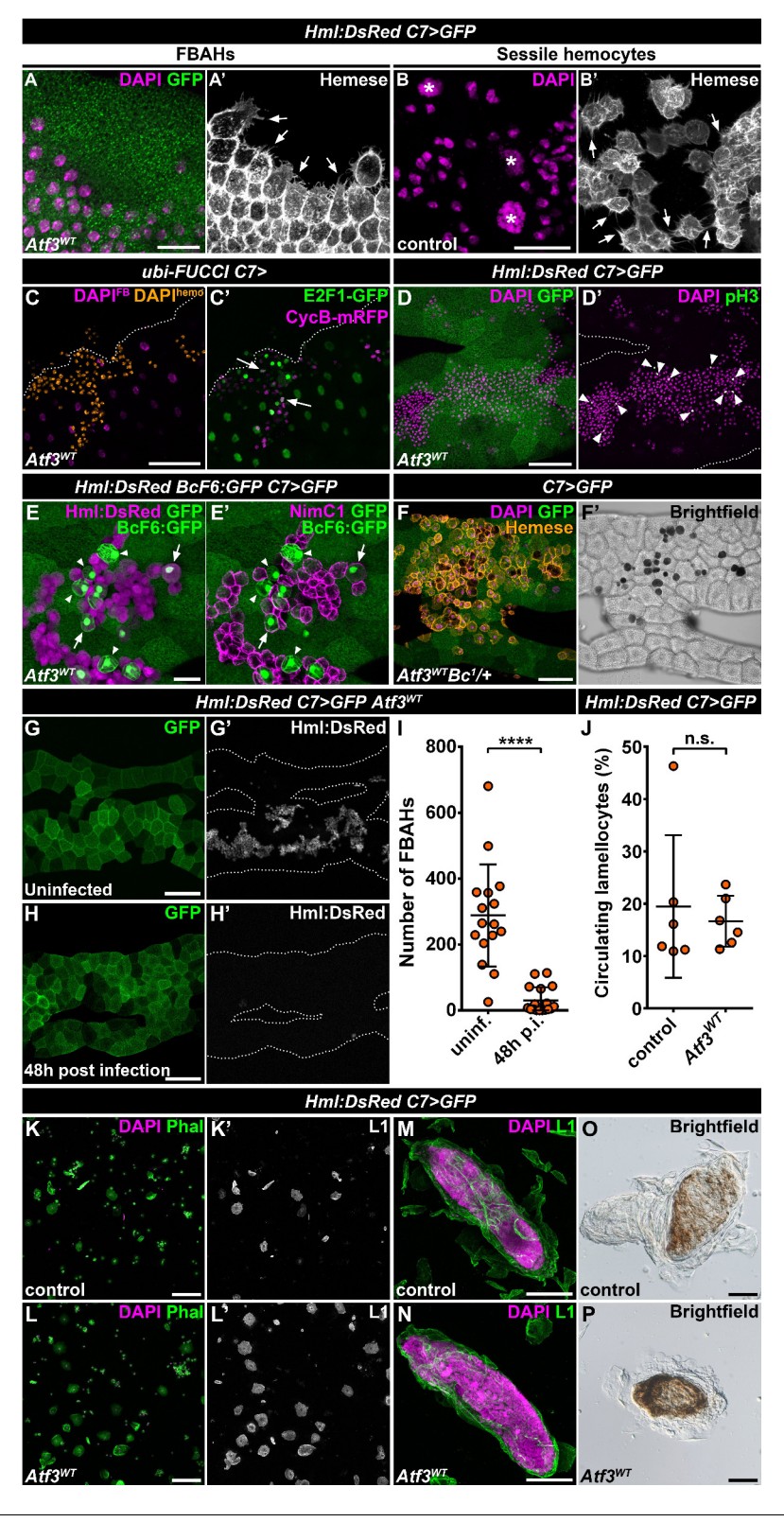

**Figure 2.** Fat body-associated hemocytes share features with sessile hemocytes. (A–B) Similar to sessile hemocytes in control larvae (B), FBAHs tightly cluster and form filopodia on the fat body surface (A). Images are depicting the fat body (A) and the epidermis (B). Transgene and GFP expression was driven by the fat body-specific *C7-GAL4* driver (A). Hemocytes were stained with pan-hemocyte anti-Hemese antibody to reveal membrane morphology. Arrows indicate filopodia emanating from hemocytes, asterisks indicate nuclei of epidermal cells (B). (C–D) The ubiquitously expressed

*Figure 2 continued on next page*

Figure 2 continued

FUCCI cell cycle reporter (C) shows FBAHs in G1 (green), S (magenta) and G2/M (white, arrows) phases of the cell cycle. Mitotic FBAHs are highlighted by pH3 staining (D, arrowheads). Nuclei were pseudocolored (see Materials and methods) to indicate hemocytes (amber) or adipocytes (magenta) (C). (E–F) Crystal cells (arrowheads), and plasmatocyte-crystal cell intermediary hemocytes (arrows) are interspersed among plasmatocytes in FBAH clusters (E). Melanized crystal cells are attached to Atf3 overexpressing fat bodies (F, black cells). Transgene and GFP expression was driven by the fat body-specific C7-GAL4 driver. Crystal cells were identified by expression of the BcF6:GFP transgene (E) or melanization due to the presence of the $Bc^1$ mutation (F). Plasmatocytes were revealed with staining against NimC1 (E'), while Hml:DsRed (E) or anti-Hemese immunostaining (F) was used to show all FBAHs. (G–I) FBAH numbers decline 48 hr after L. boulardi infection (H, I) compared to uninfected controls (G). Transgene and GFP expression was driven by the fat body-specific C7-GAL4 driver, while Hml:DsRed marks the hemocytes. Data points represent individual replicates. Unpaired nonparametric two-tailed Mann-Whitney test was used to calculate p-values. Error bars indicate SD, n ≥ 16, ****p < 0.0001 (I). (J–L) The lamellocyte differentiation 24 hr after parasitic infection (J, L) is not impaired in larvae expressing Atf3 in the fat body when compared to infected controls (J, K). Data points represent individual replicates, showing the percentage of lamellocytes (L1-positive cells) in all hemocytes (total DAPI count). Statistical significance was determined with two-tailed Student's t-test, error bars indicate SD, n = 6, n.s. = non significant (J). Images depict circulating immune cells bled from L. boulardi infected larvae. Phalloidin staining (K, L, green) labels all hemocytes, L1 staining (K', L', white) shows the lamellocytes. Transgene expression was driven by the fat body-specific C7-GAL4 driver. (M–P) Encapsulation (M) and melanization (O) of parasitic eggs are not hindered by Atf3 overexpression in the fat body (N, P). Lamellocytes surrounding the eggs 24 hr after infection were visualized with L1 staining (M, N). Brown coloration of the encapsulated eggs 48 hr following infestation indicates melanization (O, P). Nuclei were counterstained with DAPI (A–D, F, K–N). The images are projections of multiple confocal sections, fat bodies are outlined with dotted lines (C, D, G, H). Scale bars: 20 µm (A, B, E), 50 µm (C, D, F, M, N), 100 µm (G, H, K, L, O, P). See also Figure 2—source datas 1 and 2.

The online version of this article includes the following source data for figure 2:

Source data 1. Quantification of FBAH number in $C7>Atf3^{WT}$ larvae without parasitoid wasp infection and 48 hr post-infection.

Source data 2. The percentage of lamellocytes 24 hr after L. boulardi infestation in control and $C7>Atf3^{WT}$ larvae.

supplement 1C–1E and 1I) and silencing of Laminin-A, -B1 and -B2 (LanA, LanB1, LanB2) had no effect on the presence of FBAHs (Figure 3—figure supplement 1F-1I). It is important to note that similar to SPARC silencing, inhibition of Col4a1 and Trol caused marked alterations to fat body structure and impacted animal viability, as reported previously (Pastor-Pareja and Xu, 2011; Shahab et al., 2015). Surprisingly, downregulation of Mp manifested by a noticeable reduction of Mp levels (Figure 3—figure supplement 2A–2B), completely abolished hemocyte association to the Atf3 overexpressing fat body (Figure 3G-K and Figure 3—figure supplement 2C) without any adverse effect on tissue integrity, although the pericellular membrane folds with ECM material were still present (Figure 3—figure supplement 2D–2E). These results indicate that not the membrane folds, but their specific content is responsible for FBAH adhesion. To confirm this notion, we combined Mp knockdown with silencing of SPARC, which we found to exacerbate pericellular ECM accumulation (Figure 3—figure supplement 2F–2H). While the structure of $C7>Atf3^{WT}SPARC^{RNAi}Mp^{RNAi}$ fat body remained disrupted and blebbing was still present (Figure 3—figure supplement 2G) the amount of FBAHs was significantly decreased compared to $C7>Atf3^{WT}SPARC^{RNAi}$ adipose tissues (Figure 3—figure supplement 2H). Importantly, Mp knockdown in Atf3 overexpressing adipocytes resulted in restoration of the stereotypical pattern of the sessile hematopoietic pockets at the body wall (Figure 3L–O), supporting a notion that upon fat body-specific Atf3 overexpression hemocytes residing in the sessile pockets relocate and expand on the adipose tissue surface. These results demonstrate a functional link between ECM accumulation and hemocyte attachment to the fat body, and uncover Mp as a key component of this interaction.

## Similar mechanisms drive sessile hematopoietic pocket formation and FBAH adhesion

Since FBAHs phenocopy sessile hemocytes in their morphology and behavior, we asked whether they share a common mechanism of tissue attachment. While little is known about the interaction that anchors the hemocytes to the epidermis, a loss of the plasmatocyte-specific phagocytosis receptor Eater (Kocks et al., 2005) was shown to completely abolish sessile pockets, increasing the number of circulating hemocytes (Bretscher et al., 2015; Melcarne et al., 2019). To address the requirement of Eater in FBAH attachment, we generated $C7>Atf3^{WT}$ larvae homozygous for the $eater^1$ loss of function allele. Strikingly, these larvae lacked both sessile hematopoietic pockets and FBAHs (Figure 4A–E). To validate a specific requirement for Eater in the hemocytes, we first silenced

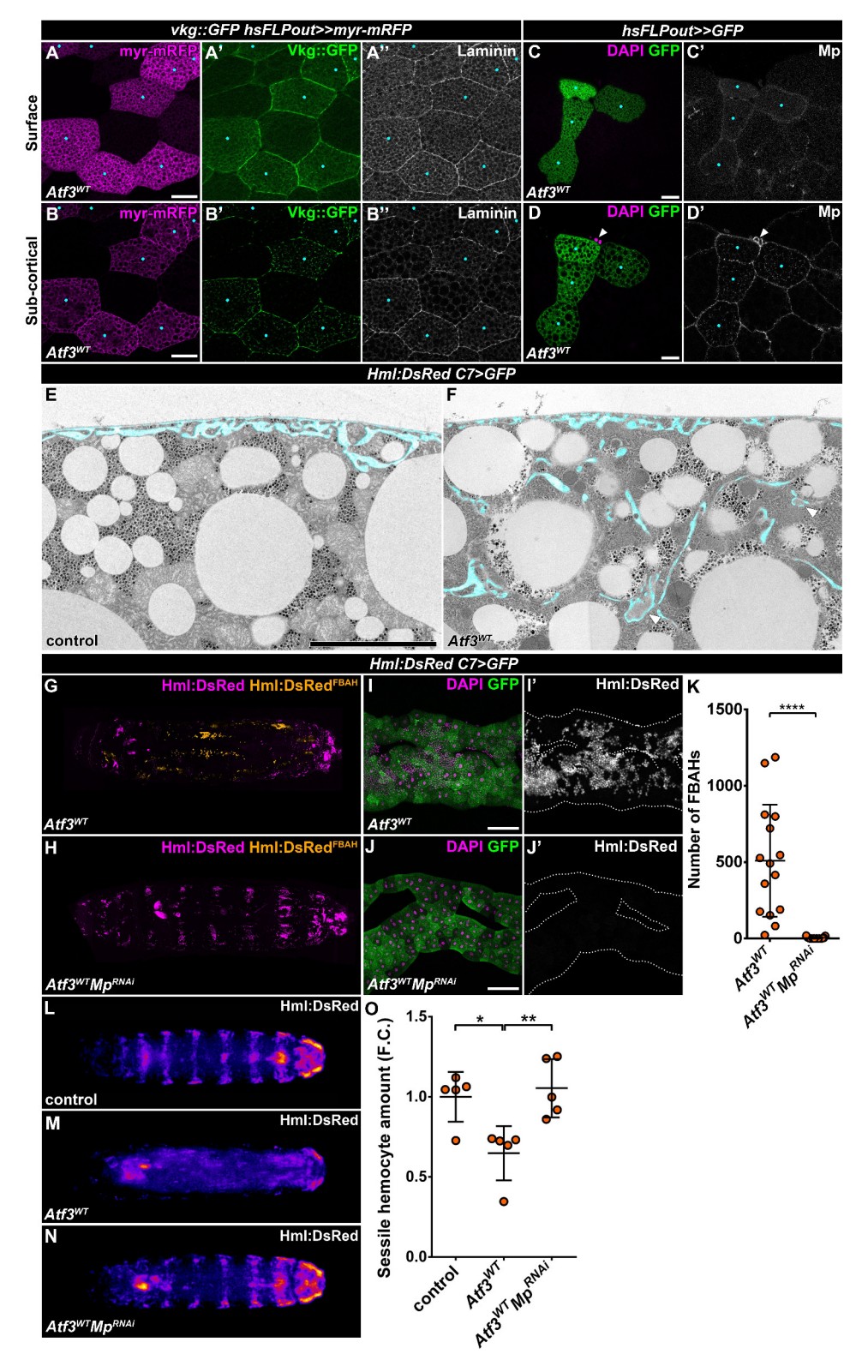

**Figure 3.** Pericellular accumulation of basement membrane components underlies hemocytes attachment to Atf3 overexpressing fat bodies. (A–D) Clonal Atf3 overexpression in the fat body leads to the enrichment of Collagen IV, Laminin and Mp on the surface of the adipocytes (A, C), and their accumulation in foci below the cell membrane (B, D). Clonal cells overexpressing Atf3 were induced with the hsFLPout technique, and are marked by the expression of *myr-mRFP* (A, B, magenta) or GFP (C, D, green). The expression of Collagen IV was visualized with the Vkg::GFP reporter, while

*Figure 3 continued on next page*

*Figure 3 continued*

Laminin and Mp expression was determined by immunostaining. The clonal adipocytes are indicated with cyan dots. The arrowhead points to hemocytes attached to the clonal cells (D). Images are single confocal planes taken at the adipose tissue surface (A, C) or ~5 µm below the tissue surface (B, D). (E–F) Transmission electron micrographs show that compared to controls (E), fat body-specific Atf3 overexpression causes the formation of cell membrane folds (F) and the entrapment of ECM material (arrowheads). The transgene expression was driven by the fat body-specific *C7-GAL4* driver. Pericellular spaces between the cell membrane and the basement membrane are colored cyan. (G–K) Knockdown of Mp in Atf3 overexpressing fat bodies abolishes the attachment of hemocytes to the adipose tissue (H, J, K) when compared to Atf3 overexpression alone (G, I, K). Whole larval images are stitches of multiple Z-projections, where FBAHs are colored amber, all other hemocytes magenta (G, H). Transgene and GFP expression was driven by the fat body-specific *C7-GAL4* driver, while *Hml:DsRed* marks the hemocytes. For whole larvae images, the localization of the DsRed signal was determined on every confocal section, and the hemocytes situated on the fat body surface were colored amber, the rest magenta (G, H). Data points represent individual replicates. Unpaired nonparametric two-tailed Mann-Whitney test was used to determine p-values, error bars indicate SD, n = 15, ****p < 0.0001 (K). (L–O) Knockdown of Mp in Atf3 overexpressing fat bodies restores the structure of the sessile hematopoietic pockets. Compared to controls (L, O), the overexpression of Atf3 in the adipose tissue disrupts the striped sessile hemocyte pattern (M, O). The pattern is restored following the simultaneous knockdown of Mp in the fat body (N, O). Transgene expression in the fat body was driven with the *C7-GAL4* driver, while the *Hml:DsRed* reporter was used to determine hemocyte location. Images represent the stereotypical DsRed pattern generated from the alignment of five individual larvae (L–N) and were used for quantification (O). Data points represent the total fluorescence intensity of four regions encompassing the four posterior-most sessile bands, which were normalized to the mean of controls (represented as 1) and shown as fold change. One-way ANOVA multiple comparison with Tukey's correction was used to determine significance, error bars indicate SD, n = 5. **p = 0.0067, *p = 0.0165. Nuclei were counterstained with DAPI (C, D, I, J). The images are single confocal sections (A–D), or represent projections of multiple confocal sections (G–J), fat bodies are outlined with dotted lines (I, J). Scale bars: 20 µm (A–D), 5 µm (E, F), 100 µm (I, J). See also *Figure 3—figure supplements 1–2* and *Figure 3—source data 1 – 2*.

The online version of this article includes the following source data and figure supplement(s) for figure 3:

**Source data 1.** Quantification of FBAH number in *C7>Atf3^{WT}* and *C7>Atf3^{WT}Mp^{RNAi}* larvae.

**Source data 2.** Quantification of FBAH number in *C7>Atf3^{WT}*, *C7>Atf3^{WT}Col4a1^{RNAi}*, *C7>Atf3^{WT}trol^{RNAi}*, *C7>Atf3^{WT}vkg^{RNAi}*, *C7>Atf3^{WT}LanA^{RNAi}*, *C7>Atf3^{WT}LanB1^{RNAi}* and *C7>Atf3^{WT}LanB2^{RNAi}* larvae.

**Source data 3.** Quantification of FBAH number in control, *C7>Atf3^{WT}*, *C7>Atf3^{WT}Mp^{RNAi}*, *C7>Atf3^{WT}SPARC^{RNAi}* and *C7>Atf3^{WT}SPARC^{RNAi}Mp^{RNAi}* larvae.

**Source data 4.** Quantification of sessile hemocyte intensity in control, *C7>Atf3^{WT}* and *C7>Atf3^{WT}Mp^{RNAi}* larvae.

**Figure supplement 1.** Knockdown of Collagen IV and Trol suppresses hemocyte adhesion to *C7>Atf3^{WT}* fat bodies.

**Figure supplement 2.** Mp is an ECM component crucial for FBAH adhesion.

its expression in *C7>Atf3^{WT}* fat bodies, which did not interfere with hemocyte-fat body interaction (*Figure 4F–G*). Next, we generated larvae which aside from *C7-GAL4* also carried the *HmlΔ-GAL4* hemocyte-specific driver (*Sinenko and Mathey-Prevot, 2004*). While the expression of Atf3 in both the fat body and the hemocytes did not markedly impact hemocyte adhesion (*Figure 4H*), simultaneous Atf3 overexpression and Eater knockdown impaired FBAH formation (*Figure 4I–J*). These data define Eater as a hemocyte-specific adhesion molecule that acts both in the context of the sessile pockets and the hemocytes on the adipose tissue surface.

The notion of a common hemocyte adhesion mechanism prompted us to test if Mp is necessary for anchoring hemocytes to the sessile pocket as it is for FBAH formation. To this end, we silenced Mp with the *a58-GAL4* (larval epidermis), *mef2-GAL4* (body wall muscles) and *elav-GAL4* (neurons) (*Galko and Krasnow, 2004*; *Lin and Goodman, 1994*; *Ranganayakulu et al., 1996*) drivers to account for the fact that the pockets reside between the larval epidermis and the body wall muscle layer, and their maintenance depends on the activity of peripheral neurons (*Makhijani et al., 2011*). While loss of Mp in the epidermal cells drastically disrupted the pattern of sessile tissue, resulting in a phenotype strikingly resembling that of *eater^1* (*Figure 4K–O*), Mp knockdown in the muscles and the neurons did not result in a similar dispersion of the sessile tissue (*Figure 4—figure supplement 1A–1D*). Interestingly, although effective in suppressing FBAHs on Atf3 overexpressing adipose tissues, knockdown of Col4a1 and Trol in the epidermis did not mimic the loss of sessile tissue structure inflicted by Mp knockdown (*Figure 4—figure supplement 1E–1F*). Furthermore, silencing Mp specifically in hemocytes had no visible impact on the integrity of the sessile hematopoietic pockets (*Figure 4—figure supplement 1G–1H*).

These results highlight a requirement for Eater and Mp to facilitate hemocyte-basement membrane interactions both in the case of the naturally occurring sessile hematopoietic pockets and the de novo hematopoietic compartment on Atf3 overexpressing fat bodies.

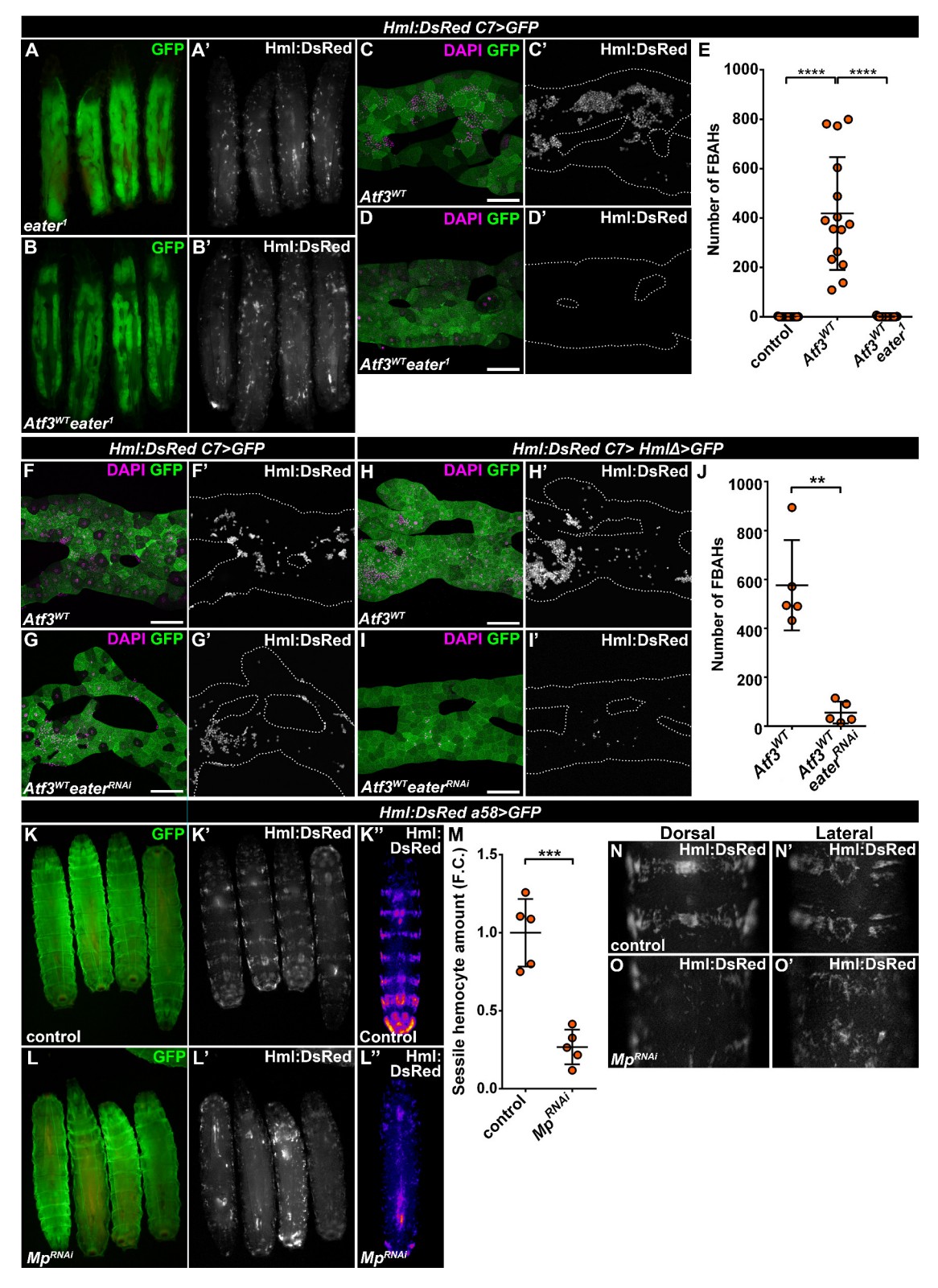

**Figure 4.** Mp and Eater control hemocyte attachment to the sessile hematopoietic pockets and on the surface of Atf3 overexpressing fat bodies. (A–E) Eater loss abrogates the association of hemocytes both to the sessile pockets (A), and to Atf3 overexpressing fat bodies (B, D, E) compared to Atf3 overexpression alone (C, E). Transgene and GFP expression was driven by the fat-body-specific *C7-GAL4* driver, while *Hml:DsRed* marks the hemocytes. Data points represent individual replicates. Nonparametric one-way Kruskal-Wallis test with Dunn's multiple comparison was used to

*Figure 4 continued on next page*

Figure 4 continued

determine significance. Error bars indicate SD, n = 15. ****adjusted p < 0.0001 (E). (F–J) The knockdown of Eater in the Atf3 overexpressing fat body does not noticeably influence the association of hemocytes (G) compared to Atf3 overexpression alone (F). The combined expression of Atf3 in the fat body and the hemocytes does not markedly alter FBAH cluster formation (H), while simultaneous Eater knockdown suppresses hemocyte attachment (I, J), indicating hemocyte-specific requirement for Eater function. Transgene and GFP expression was driven either by the combination of the fat-body-specific *C7-GAL4* driver and the hemocyte-specific *HmlΔ-GAL4* (H–J), or with the *C7-GAL4* alone (F, G), while *Hml:DsRed* marks the hemocytes. Data points represent individual replicates. Significance was determined by unpaired nonparametric two-tailed Mann-Whitney test, error bars represent SD, n = 5, **p = 0.0097 (J). (K–L) Knockdown of Mp in epidermal cells (L) disrupts the stereotypical banded sessile hemocyte pattern (K). Transgene and GFP expression was driven by the epidermis-specific *a58-GAL4* driver, while *Hml:DsRed* marks the hemocytes. Images represent individual larvae (K, K', L, L'), or the stereotypical DsRed pattern generated from the alignment of five individual larvae (K'', L''). (M) Sessile hemocyte amounts significantly decrease following epidermal knockdown of Mp. Transgene expression was driven by the *a58-GAL4* driver, while *Hml:DsRed* marks the hemocytes. Data points represent the total fluorescence intensity of four regions encompassing the four posterior-most sessile bands, which were normalized to the mean of controls (represented as 1) and shown as fold change. Statistical significance was determined with two-tailed student's t-test, error bars indicate SD, n = 5. ****p < 0.0001. (N–O) Compared to controls (N), the structure of both the dorsal stripe and lateral patches of the sessile hematopoietic tissue is disrupted upon knockdown of Mp in the epidermis (O). Note that hemocyte accumulation on the lateral side of larvae with epidermal-specific Mp knockdown is likely the consequence of decreased hemolymph flow due to the immobilization process (O'). Transgene expression was driven by the *a58-GAL4* driver, while *Hml:DsRed* marks the hemocytes. Images depict the dorsal (N, O) and lateral (N', O') views of the A5-A6 larval segments from the same larvae. Tissues were counterstained with DAPI (C, D, F–I). Images are maximum projections of multiple confocal sections (C, D, F–I). Fat bodies are outlined with dotted lines (C, D, F–I). Scale bars: 100 μm (C, D, F–I). See also *Figure 4—figure supplement 1* and *Figure 4—source datas 1–3*.

The online version of this article includes the following source data and figure supplement(s) for figure 4:

**Source data 1.** Quantification of FBAH number in control, *C7>Atf3^{WT}* and *C7>Atf3^{WT}eater^1* larvae.

**Source data 2.** Quantification of FBAH number in *C7>Hml>Atf3^{WT}* and *C7>Hml>Atf3^{WT}eater^{RNAi}* larvae.

**Source data 3.** Quantification of sessile hemocyte intensity in control and *a58 >Mp^{RNAi}* larvae.

**Figure supplement 1.** Col4a1 and Trol knockdown in the epidermis or Mp knockdown in the hemocytes, the muscles and the neurons do not affect the epidermal sessile hematopoietic compartment.

## Immune cell-tissue attachment depends on the interaction of Multiplexin and Eater

The *Drosophila* Mp consists of an N-terminal Thrombospondin-like domain, followed by a Collagen triple helix domain, an NC1 trimerization domain and the C-terminal Endostatin-domain, all of which are present in its mammalian counterparts, Collagens XV and XVIII (*Heljasvaara et al., 2017*). Mp expression initiates during late embryonic development, mostly in the heart tube and the central nervous system, and is needed for motoaxonal pathfinding (*Harpaz et al., 2013*; *Meyer and Moussian, 2009*). These traits are distinct from the ubiquitous Col4a1 and Vkg, suggesting that Mp-containing basement membranes may have specific functions in cell adhesion and migration. Given the fact that hemocyte attachment to the epidermis as well as to *C7>Atf3^{WT}* fat bodies was dependent on Mp, we asked whether its presence in the basement membrane may be sufficient to promote immune cell attachment. To this end, we generated a C-terminally GFP-tagged Mp transgene (*UAS-Mp::GFP*) and overexpressed it with the *C7-GAL4* driver in the adipose tissue. Compared to control, Mp levels markedly increased in the basement membrane of *C7>Mp::GFP* fat body (*Figure 5A–B*) as determined by immunostaining with a Mp-specific antibody (*Harpaz et al., 2013*). However, GFP signal was restricted to the adipocyte cytoplasm (*Figure 5—figure supplement 1A*), suggesting that the C-terminal end of Mp undergoes proteolytic cleavage. Surprisingly, Mp::GFP overexpression in the fat body did not induce hemocyte adhesion (*Figure 5A–B*). Instead, it led to the complete dispersal of the sessile hematopoietic pockets, phenocopying homozygous *eater^1* mutants (*Figure 5C–E*, compare to *Figure 4A*). In addition, circulating hemocyte numbers in *C7>Mp::GFP* larvae significantly increased compared to controls, reaching similar levels as in *eater^1* mutants (*Figure 5F*). Immunoblots from the cell free hemolymph revealed a notable increase of the Mp protein in the circulation of *C7>Mp::GFP* larvae relative to controls (*Figure 5G*). These results indicate that while fat-body-produced Mp incorporates into the adipocyte ECM, it is also released into the hemolymph where it may interact with hemocytes and interfere with their binding to the basement membrane due to a saturation effect.

To circumvent the hemolymph overload, we induced Mp::GFP expression only in clones, which resulted in local Mp accumulation over the targeted adipocytes and in their intercellular spaces (*Figure 6A–E*). Strikingly, clonal adipocytes overexpressing Mp::GFP were surrounded by

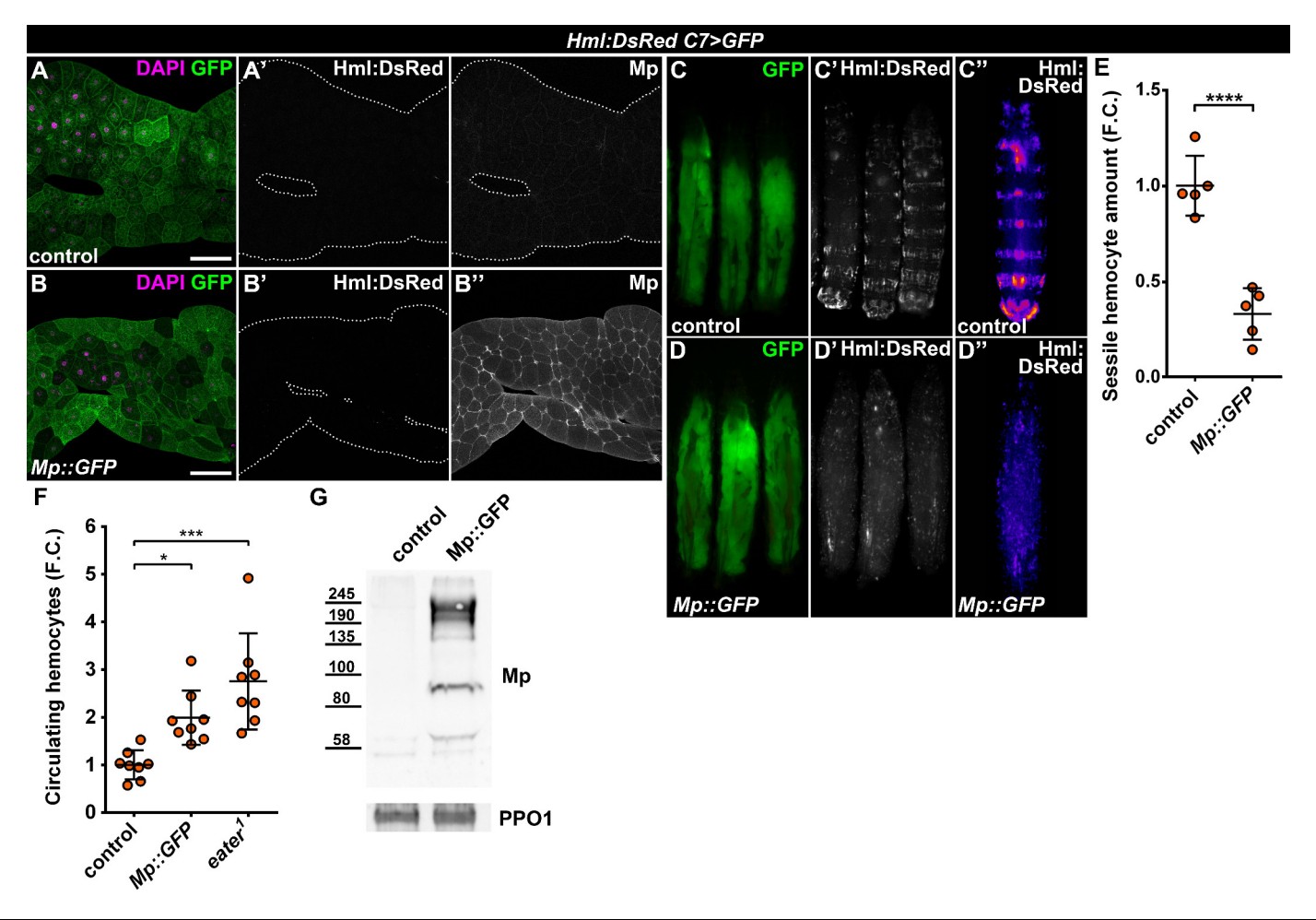

**Figure 5.** Fat body-wide Mp overexpression causes detachment of the sessile hemocytes. (**A–B**) Similar to controls (**A**) fat bodies overexpressing Mp::GFP do not attract hemocytes (**B**), even though Mp integrates into the basement membrane of the adipose tissue (**A''**, **B''**). Transgene and GFP expression was driven by the fat-body-specific *C7-GAL4* driver, while *Hml:DsRed* marks the hemocytes. The expression of Mp was determined with immunostaining. Fat bodies are outlined with dotted lines. Nuclei were counterstained with DAPI. Scale bars: 100 μm. (**C–E**) Fat body-specific Mp::GFP expression disrupts the segmentally organized sessile hematopoietic compartment (**D**) as observed in controls (**C**). Images represent individual larvae (**C, C', D, D'**) or the stereotypical Hml:DsRed pattern generated from the alignment of five individual larvae (**C'', D''**). Data points represent the total fluorescence intensity of four regions encompassing the four posterior-most sessile bands, which were normalized to the mean of controls (represented as 1) and shown as fold change. Statistical significance was determined with two-tailed Student's t-test, error bars indicate SD, n = 5, ****p < 0.0001 (**E**). (**F**) Sessile hemocyte detachment following fat body-wide overexpression of Mp::GFP coincides with the elevation of circulating hemocyte numbers similar to *eater* deficiency. Data points represent individual replicates, which were normalized to control mean (represented as 1). Nonparametric one-way Kruskal-Wallis test with Dunn's multiple comparison was used to determine significance, error bars indicate SD, n = 8, ***adjusted p = 0.0003, *adjusted p = 0.0175. (**G**) Mp levels increase in the circulation upon fat body-specific overexpression. Immunoblot against Mp shows multiple bands in cell-free hemolymph extracts, indicating extensive post-translational processing. Molecular weights (in kDa) are shown. Prophenoloxidase 1 (PPO1) served as a loading control. The expression of transgenes and GFP was driven with the fat body-specific *C7-GAL4* driver, while hemocytes were recognized based on the expression of the *Hml:DsRed* reporter (**A–G**). See also *Figure 5—figure supplement 1* and *Figure 1—source datas 1–2*. The online version of this article includes the following source data and figure supplement(s) for figure 5:

**Source data 1.** Quantification of sessile hemocyte intensity in control and *C7>Mp::GFP* larvae.
**Source data 2.** Circulating hemocyte counts from control, Mp::GFP overexpressing and *eater[1]* mutant larvae.
**Figure supplement 1.** The Mp::GFP transgenic protein is incorporated into the ECM without the GFP tag.

hemocytes, which displayed previously established characteristics of FBAHs, namely tight clustering and projection of filopodia and lamellipodia, the ability to undergo mitosis on the tissue surface and the presence of crystal cells (*Figure 6C–E*). Moreover, Mp::GFP expression in the pouch region of the wing disc using the *nubbin-Gal4, UAS-mCherry* driver (*nub>mCherry*) promoted hemocyte

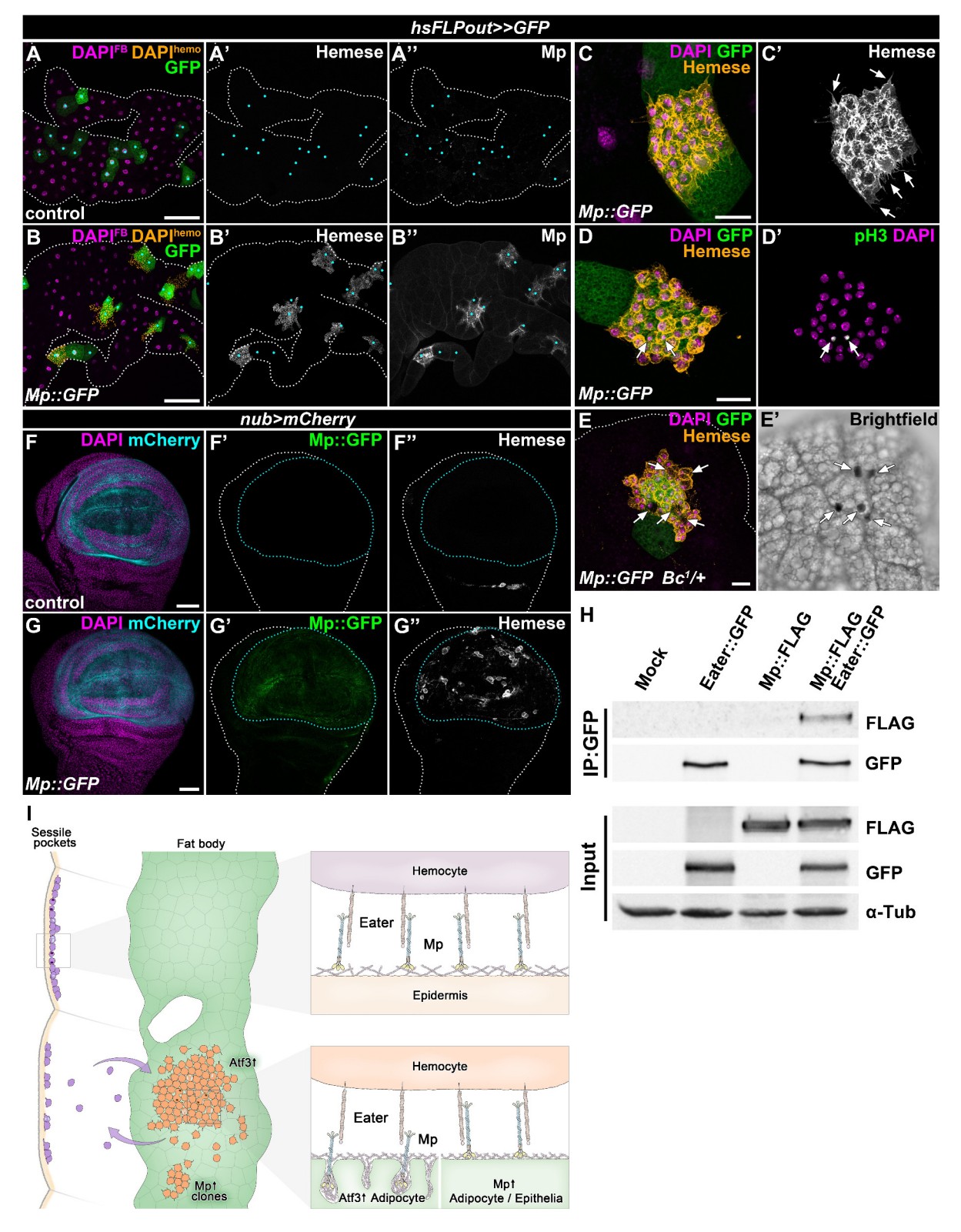

**Figure 6.** The interaction of Mp and Eater underlies hemocyte attachment to tissue surfaces. (A–B) In contrast to expression in the whole adipose tissue, clonal overexpression of Mp::GFP in the fat body attracts hemocytes and causes local incorporation of Mp into adipocyte basement membrane, (B) compared to controls (A). Heat-shock induced FLPout clones were distinguished based on their expression of GFP. Clonal adipocytes are indicated with cyan dots. Nuclei were pseudocolored amber to indicate FBAH nuclei, and magenta to show adipocyte nuclei. (C–D) Hemocytes attached to the

*Figure 6 continued on next page*

*Figure 6 continued*

surface of Mp::GFP overexpressing adipocyte clones (**C, D**) cluster tightly together and extend filopodia (**C'**, arrows), and some undergo cell division (**D'**). Immunostaining against Hemese visualizes hemocytes (C, D, amber, C', white), phospho-histone H3 staining shows mitotic nuclei (D', white, indicated with arrows). (**E**) Crystal cells (indicated with arrows) are present on the surface of Mp::GFP overexpressing adipocyte clones (**E**). Immunostaining against Hemese was used to visualize hemocytes, and melanized crystal cells can be identified due to the presence of the *Bc*[1] mutation (E', black cells). (**F–G**) While in control wing discs no hemocytes can be observed on the basal side of the wing pouch (**F**), overexpression of Mp::GFP in this domain using the *nub-GAL4, UAS-mCherry* driver (F, G, cyan, outlined with cyan dotted lines) is sufficient to cause hemocyte attachment (**G**). The hemocytes were visualized with immunostaining against Hemese. Images represent projections of multiple confocal sections from the basal side of the wing disc. (**H**) Mp::FLAG co-precipitates with Eater::GFP from *Drosophila* S2 cells lysates. The GFP-tagged Eater served as the bait (IP:GFP). Eater and Mp proteins were detected with the anti-GFP and anti-FLAG tag-specific antibodies. The lower panel shows input extracts with α-Tubulin serving as a loading control. (**I**) Adipocyte-specific overexpression of Atf3 redirects hemocytes (orange) to the fat body surface (green) from the sessile hematopoietic pockets (purple), where they proliferate (cells with two nuclei) and trans-differentiate into crystal cells (cyan) and can detach from upon immune challenge, similar to their natural hematopoietic environment. The presence of Mp in the basement membrane promotes hemocyte attachment both in the sessile compartment and on the fat body surface upon Atf3 or clonal Mp expression through its interaction with the phagocytosis receptor Eater (right panels). Tissues were counterstained with DAPI (**A–G**). Images are projections of multiple confocal sections (**A–G**). Fat bodies (A, B, E) or wing discs (F) are outlined with white dotted lines. Scale bars: 100 µm (A, B), 20 µm (C–E), 50 µm (F, G). See also *Figure 6— figure supplement 1*.

The online version of this article includes the following figure supplement(s) for figure 6:

**Figure supplement 1.** The overexpression of Mp:GFP in the wing pouch region does not induce apoptosis.

adhesion to this specific domain (*Figure 6F–G*). Of note, this hemocyte epithelial tissue association was not provoked by apoptosis of imaginal cells (*Figure 6—figure supplement 1A–1B*). Together these results demonstrate that Mp is not only necessary but also sufficient to facilitate hemocyte attachment to tissue surfaces.

The necessity of Eater and Mp to secure the immune cells to the basement membrane and the results from Mp overexpression experiments prompted us to test if the two proteins might physically interact. To this end, we performed co-immunoprecipitation experiments with tagged Eater::GFP and Mp::FLAG proteins in S2 cells, and found that the two proteins indeed co-precipitated (*Figure 6H*). While we cannot exclude that the binding might be indirect, these results present an argument that Mp acts as an interacting partner of Eater on the basement membranes of *C7>Atf3$^{WT}$* fat bodies, and on the epidermal surface covering the sessile hematopoietic pockets (*Figure 6I*).

## Discussion

The hematopoietic microenvironments are essential for immune cell development and hematopoietic homeostasis by providing molecular cues and physical interactions that control HSC and progenitor cell localization, maintenance and differentiation. In contrast, emerging evidence suggest that niche alterations can drive premature hematopoietic aging and malignancies (*Ho et al., 2019*; *Walkley et al., 2007a*; *Walkley et al., 2007b*). Fundamental principles of hematopoiesis and hematopoietic niche formation show similarities across phyla (*Martinez-Agosto et al., 2007*). In this study, we employed the *Drosophila melanogaster* model to gain a mechanistic understanding of how hematopoietic microenvironments arise. We show that Atf3 overexpression in adipocytes promotes the formation of a de novo hematopoietic compartment on the adipose tissue surface at the expense of the naturally occurring sessile hematopoietic cell pool (*Figure 6I*). The fat body-associated hemocytes showed round/spherical morphology, and projected short filopodia and lamellipodia, which are characteristic to the sessile hemocytes (*Lanot et al., 2001*, *Figure 2B*) and hemocytes adhering to Laminin-coated substrates (*Sampson and Williams, 2012*), but are distinct from the elongated phenotype of the migratory embryonic macrophages and the pupal hemocytes assisting tissue remodeling (*Evans and Wood, 2011*; *Sampson et al., 2013*). The characteristic morphology and the absence of tissue debris-phagocytosis and encapsulation response suggest that FBAHs are in a naïve homeostatic state, not engaged in immune response. Moreover, similar to unchallenged sessile hemocytes, FBAHs proliferate and differentiate in situ on the adipose tissue surface. Although hemocytes also proliferate in the circulation, the division rate of those in the sessile pockets was shown to be higher due to the stimulating effects of Activin-β secreted by peripheral sensory neurons (*Makhijani et al., 2017*; *Makhijani et al., 2011*). Since the fat body lacks the innervation of the

epidermal cells, the proliferation of FBAHs may be part of a hemocyte-autonomous developmental program, which is supported by the physical adhesion to the tissue and cell-cell contacts between attached hemocytes. Such contacts appear instrumental in the trans-differentiation of plasmatocytes to crystal cells in the sessile pockets, which requires Serrate-Notch juxtacrine signaling among the clustered cells (*Leitão and Sucena, 2015*). Similarly, FBAH clusters contained fully and partially differentiated crystal cells intermingled with the hemocytes, which suggests that plasmatocyte-crystal cell trans-differentiation is cluster-dependent and does not require signals secreted by peripheral neurons.

Importantly, the simultaneous appearance of the FBAHs and a decline of the sessile hemocytes did not significantly impact the amount of circulating blood cells. This may indicate that FBAHs are dynamic, entering and leaving the circulation at a comparable rate to sessile hemocytes (*Honti et al., 2010*; *Makhijani et al., 2011*; *Welman et al., 2010*). The dynamic nature of FBAHs is further underlined by the fact that they detached after parasitic wasp infestation, mimicking the described behavior of the sessile population (*Márkus et al., 2009*; *Vanha-Aho et al., 2015*).

The analysis of FBAHs provided us with an avenue to explore the mechanistic underpinnings of hemocyte-tissue interactions, and highlighted the importance of basement membrane proteins in the formation of *Drosophila* hematopoietic tissues. We identified Mp, the *Drosophila* ortholog of Collagens XV/XVIII as a necessary ECM protein for hemocyte attachment to tissue surfaces. We show that Mp loss not only blocked hemocyte attachment to the Atf3 overexpressing fat body, but also restored the integrity of the sessile pockets, while its epidermal knockdown caused sessile tissue disintegration. It is important to note that the formation of FBAHs, but not the sessile hematopoietic pockets, was abolished also by inhibiting Col4a1 and Trol. While these ECM components may directly facilitate hemocyte attachment, it is more likely that they are required to form the general lattice structure of the basement membrane to which Mp anchors. Supporting this argument, when ECM accumulation in the fat body was exacerbated by inhibiting SPARC, simultaneous Mp loss abolished FBAH clusters without restoring the fat body integrity. Thus, unlike the ubiquitously present ECM components Collagen IV, Laminins or Trol (*Pastor-Pareja and Xu, 2011*) the expression of Mp is restricted to tissues where it locally incorporates into the basement membrane to facilitate cell adhesion (*Harpaz et al., 2013*; *Meyer and Moussian, 2009*).

Importantly, we show that the clonal gain of Mp induced localized FBAH clusters very similar to those found on Atf3-expressing adipocytes, demonstrating that Mp is not only required but also sufficient to promote hemocyte adhesion. Furthermore, Mp expression in the wing pouch also attracted hemocytes to the wing disc surface, which suggests that this phenomenon is not restricted to the adipose tissue. Surprisingly, Mp overexpression in the entire fat body did not attract hemocytes. Instead, it resulted in the complete loss of the sessile population. We propose that the increased Mp levels in the hemolymph compete with the basement membrane-incorporated protein for the binding of its receptor on the hemocytes preventing their association with the epidermal or fat body ECMs. These results also suggest that both the tissue specificity and the levels of endogenous Mp expression are tightly regulated. In the context of the sessile hematopoietic pockets, this could mean that low amounts of Mp in the epidermal basement membrane are sufficient to anchor hemocytes through a specific and strong interaction between Mp and its receptor on the hemocytes.

The de novo hematopoietic tissue model and the biochemical assay from S2 cells imply that this hemocyte-specific binding partner is the scavenger receptor Eater. Although the phagocytic properties of Eater have been extensively characterized (*Chung and Kocks, 2011*; *Kocks et al., 2005*; *Melcarne et al., 2019*), its requirement for the formation of hematopoietic pockets was only recently established (*Bretscher et al., 2015*; *Melcarne et al., 2019*). We find that Eater is not only essential to maintain the sessile hemocytes but also needed for FBAH formation, further underlining the idea of common hemocyte attachment mechanisms (*Figure 6I*).

Interestingly, the mammalian counterpart of Mp, Collagen XV is restricted to the cardiac and skeletal muscles (*Hägg et al., 1997*) while its other ortholog Collagen XVIII is ubiquitously expressed throughout development (*Miosge et al., 2003*). Both ColXV and ColXVIII undergo a multitude of posttranslational modifications, which include the addition of chondroitin- or heparan-sulfate sidechains (*Dong et al., 2003*), and extensive protein cleavage which produces Endostatin, an anti-angiogenic peptide that is widely studied because of its link to cell migration, proliferation and

tumorigenesis (*O'Reilly et al., 1997*; *Walia et al., 2015*). Similarly, Mp was found to be modified by Glycosaminoglycan (GAG) chains, and the overexpression of its N- and C-terminal domains had distinct effects on the motoaxon guidance (*Meyer and Moussian, 2009*; *Momota et al., 2011*). The hemocyte-specific counterpart of Mp, Eater, on the other hand, is a member of the Nimrod superfamily of transmembrane phagocytosis receptors that contain multiple EGF-like domains (*Kocks et al., 2005*; *Somogyi et al., 2008*). While Nimrod family members have no direct orthologs in mammals, multiple EGF-like domain-containing transmembrane proteins such as MEGF10 have been associated with cell adhesion and migration (*Suzuki and Nakayama, 2007*). Since Mp is extensively processed upon its secretion to the extracellular space, it is tempting to speculate that its interaction with Eater may be due to particular posttranslational modifications, such as chondroitin sulfate addition or proteolytic cleavage (*Meyer and Moussian, 2009*; *Momota et al., 2011*), which sets it apart from the ubiquitously present ECM proteins in *Drosophila*.

# Materials and methods

## Key resources table

| Reagent type (species) or resource | Designation | Source or reference | Identifiers | Additional information |
|---|---|---|---|---|
| Strain, strain background (*Drosophila melanogaster*) | $w^{1118}$ | BDSC | RRID:BDSC_3605 | |
| Strain, strain background (*Drosophila melanogaster*) | w; C7-GAL4 | *Rynes et al., 2012* | | |
| Strain, strain background (*Drosophila melanogaster*) | w; UAS-Atf3$^{WT}$ | *Sekyrova et al., 2010* | | |
| Strain, strain background (*Drosophila melanogaster*) | w; Hml:DsRed | *Makhijani et al., 2011* | | |
| Strain, strain background (*Drosophila melanogaster*) | w; hsFLP, act>y$^+$>GAL4, UAS-GFP | *Sekyrova et al., 2010* | | |
| Strain, strain background (*Drosophila melanogaster*) | w; hsFLP, act>y$^+$>GAL4, UAS-myr.mRFP | *Sekyrova et al., 2010* | | |
| Strain, strain background (*Drosophila melanogaster*) | w;; Ubi-GFP.E2f1$^{1-230}$, Ubi-mRFP1.NLS.CycB$^{1-266}$ | *Zielke et al., 2014* | RRID:BDSC_55124 | |
| Strain, strain background (*Drosophila melanogaster*) | w;; BcF6:GFP | *Tokusumi et al., 2009* | | |
| Strain, strain background (*Drosophila melanogaster*) | w; Bc$^1$ | *Rizki et al., 1980* | | |
| Strain, strain background (*Drosophila melanogaster*) | w; P[PTT-un1]vkgG454 | *Morin et al., 2001* | DGRC 11069 | |

*Continued on next page*

Continued

| Reagent type (species) or resource | Designation | Source or reference | Identifiers | Additional information |
|---|---|---|---|---|
| Strain, strain background (*Drosophila melanogaster*) | *w; Mp^RNAi* | VDRC | v38189 | |
| Strain, strain background (*Drosophila melanogaster*) | *w; Mp^RNAi* II | VDRC | v35431 | |
| Strain, strain background (*Drosophila melanogaster*) | *w; Cg25C^RNAi* | VDRC | v28369 | |
| Strain, strain background (*Drosophila melanogaster*) | *w, trol^RNAi* | VDRC | v22642 | |
| Strain, strain background (*Drosophila melanogaster*) | *w; vkg^RNAi* | VDRC | v16986 | |
| Strain, strain background (*Drosophila melanogaster*) | *w; LanA^RNAi* | VDRC | v18873 | |
| Strain, strain background (*Drosophila melanogaster*) | *w; LanB1^RNAi* | VDRC | v23119 | |
| Strain, strain background (*Drosophila melanogaster*) | *w; LanB2^RNAi* | VDRC | v42559 | |
| Strain, strain background (*Drosophila melanogaster*) | *w; SPARC^RNAi* | VDRC | v16677 | |
| Strain, strain background (*Drosophila melanogaster*) | *w;; a58-GAL4* | **Galko and Krasnow, 2004** | | |
| Strain, strain background (*Drosophila melanogaster*) | *w;; mef2-GAL4* | **Ranganayakulu et al., 1996** | RRID:BDSC_27390 | |
| Strain, strain background (*Drosophila melanogaster*) | *w, elav-GAL4* | **Lin and Goodman, 1994** | RRID:BDRSC_458 | |
| Strain, strain background (*Drosophila melanogaster*) | *eater^1* | **Bretscher et al., 2015** | RRID:BDSC_68388 | |
| Strain, strain background (*Drosophila melanogaster*) | *w; HmlΔ-GAL4* | **Sinenko and Mathey-Prevot, 2004** | RRID:BDSC_30139 | |
| Strain, strain background (*Drosophila melanogaster*) | *w; nub-GAL4, UAS-mCherry* | BDSC | RRID:BDSC_63148 | |

*Continued*

| Reagent type (species) or resource | Designation | Source or reference | Identifiers | Additional information |
|---|---|---|---|---|
| Strain, strain background (*Drosophila melanogaster*) | *w;; UAS-eater*[RNAi] | VDRC | v4301 | |
| Strain, strain background (*Drosophila melanogaster*) | *w;; UAS-Mp::GFP* | This study | | |
| Cell line (*Drosophila melanogaster*) | Schneider 2 (S2) cells | *Drosophila* Genomic Resource Center | RRID:CVCL_Z992 | |
| Recombinant DNA reagent | pENTR4-dual | Thermo Fisher | A10465 | |
| Recombinant DNA reagent | pTWG | DGRC | 1076 | |
| Recombinant DNA reagent | pTWF | DGRC | 1116 | |
| Recombinant DNA reagent | pTWG-Mp | This study | | Used to generate UAS-Mp::GFP *Drosophila* line |
| Transfected construct (*Drosophila melanogaster*) | pTWF-Mp | This study | | Used to transfect S2 cells |
| Transfected construct (*Drosophila melanogaster*) | pTWG-Eater | This study | | Used to transfect S2 cells |
| Transfected construct (*Drosophila melanogaster*) | pWA-GAL4 | Oda and Tsukita, 1999 | | Used to transfect S2 cells |
| Antibody | Anti-Hemese (mouse monoclonal) | *Kurucz et al., 2003* (I. Ando) | H2 | IF (1:100) |
| Antibody | Anti-L1 (mouse monoclonal) | *Kurucz et al., 2007b* (I. Ando) | H10 | IF (1:100) |
| Antibody | Anti-NimrodC1 (mouse monoclonal) | *Kurucz et al., 2007a* (I. Ando) | N1+N47 | IF (1:100) |
| Antibody | Anti-Laminin (rabbit polyclonal) | Abcam | ab47651, RRID:AB_880659 | IF (1:500) |
| Antibody | Anti-alpha-Tubulin (mouse monoclonal) | DSHB | AA4.3, RRID:AB_579593 | IF (1:200), WB (1:1000) |
| Antibody | Anti-phospho-histone H3 (rabbit polyclonal) | Cell Signaling | Cat# 9701, RRID:AB_331535 | IF (1:500) |
| Antibody | Anti-cleaved-Dcp-1 (rabbit polyclonal) | Cell Signaling | Cat# 9578, RRID:AB_2721060 | IF (1:500) |
| Antibody | Anti-Endostatin (rat polyclonal) | *Harpaz et al., 2013* (T. Volk) | | IF (1:200), WB (1:1000) |
| Antibody | Anti-FLAG M2 (mouse monoclonal) | Sigma-Aldrich | F1804, RRID:AB_439685 | WB (1:1000) |
| Antibody | Anti-GFP (rabbit polyclonal) | Acris | TP401, RRID:AB_2313770 | WB (1:5000) |
| Antibody | Anti-PPO1 (rabbit polyclonal) | *Jiang et al., 1997* (M. Kanost) | | WB (1:750) |

*Continued on next page*

*Continued*

| Reagent type (species) or resource | Designation | Source or reference | Identifiers | Additional information |
|---|---|---|---|---|
| Antibody | Anti-mouse IgG Cy3 (donkey polyclonal) | Jackson Immuno Research | 715-165-1511 | IF (1:500) |
| Antibody | Anti-mouse IgG Cy5 (donkey polyclonal) | Jackson Immuno Research | 715-175-1510 | IF (1:500) |
| Antibody | Anti-rabbit IgG Cy3 (donkey polyclonal) | Jackson Immuno Research | 711-165-152 | IF (1:500) |
| Antibody | Anti-rabbit IgG Cy5 (donkey polyclonal) | Jackson Immuno Research | 711-175-152 | IF (1:500) |
| Antibody | Anti-rat IgG Cy5 (donkey polyclonal) | Jackson Immuno Research | 712-175-153 | IF (1:500) |
| Antibody | Anti-mouse IgG HRP (donkey polyclonal) | Jackson Immuno Research | 715-035-150 | WB (1:5000) |
| Antibody | Anti-rabbit IgG HRP (donkey polyclonal) | Jackson Immuno Research | 711-035-152 | WB (1:5000) |
| Antibody | Anti-rat IgG HRP (donkey polyclonal) | Jackson Immuno Research | 712-035-153 | WB (1:5000) |
| Chemical compound, drug | 4′,6-Diamidine-2′-phenylindole (DAPI) | Carl Roth GmBH. | 6335,1 | 1 mg/ml |
| Chemical compound, drug | N-Phenylthiourea | Sigma-Aldrich | P7629 | 0.01% w/V |
| Software, algorithm | FIJI | *Schindelin et al., 2012* | http://fiji.sc, RRID:SCR_003070 | |
| Software, algorithm | GraphPad Prism 6 | GraphPad | RRID:SCR_002798 | |
| Software, algorithm | Photoshop CS5.5 | Adobe Systems, Inc | RRID:SCR_014199 | |
| Software, algorithm | FluoView FV-10ASW | Olympus | RRID:SCR_014215 | |
| Software, algorithm | cellSens standard v1.11 | Olympus | RRID:SCR_014551 | |
| Software, algorithm | CellProfiler | *Kamentsky et al., 2011* | RRID:SCR_007358 | |
| Other reagent | Phalloidin-Alexa 488 | Molecular Probes | A12379 | Used for F-actin staining |
| Other reagent | Dabco-Mowiol | Sigma-Aldrich | D27802,81381 | Mounting medium |
| Other reagent | TransIT-Insect Reagent | Mirus | MIR6100 | Tranfection reagent |
| Other reagent | GFP-trap beads | Chromotek | gtma-20, RRID:AB_2631406 | GFP trap beads for co-immunoprecipitation |
| Commercial assay or kit | Gateway LR Clonase II | Thermo Fisher | 11791–020 | Gateway clonase for entry-destination (LR) recombination |

## *Drosophila* husbandry

Flies were kept at 25°C on a diet consisting of 0.8% (wt/vol) agar, 8% cornmeal, 1% soy meal, 1.8% dry yeast, 8% malt extract, and 2.2% sugar beet syrup supplemented with 0.625% propionic acid and 0.15% Methylparaben (Sigma-Aldrich). In standard crosses, 10 virgins were crossed with five males. The crosses were flipped daily, and the larvae were analyzed 6 days after egg laying (AEL). Driver lines crossed to $w^{1118}$ served as controls.

Clones overexpressing specific transgene were generated with the FLPout technique utilizing heat shock-induced FLP expression which removes an FRT-flanked stop cassette separating a constitutive *Act5C* promoter from the GAL4 coding sequence (*hsFLP, Act5C>y⁺>GAL4*). Clonal cells were marked by the expression of *UAS-GFP* or *UAS-myr-mRFP*. To generate adipocyte clones, first instar larvae (24 hr AEL) were heat-shocked in a 37°C water bath for 20 min, and were afterwards kept at 25°C until processing on day six AEL.

### *Drosophila melanogaster* lines

The following *Drosophila melanogaster* strains were used: *w¹¹¹⁸* (BDSC; RRID:BDSC_3605), *w; Hml:DsRed* (*Makhijani et al., 2011*), *w; C7-GAL4* (*Rynes et al., 2012*), *w; C7-GAL4, UAS-Atf3^WT^* (*Rynes et al., 2012*), *w; C7-GAL4, UAS-GFP, Hml:DsRed* (recombined in this study), *w; C7-GAL4, UAS-GFP, Hml:DsRed, UAS-Atf3^WT^* (recombined in this study), *w; nub-GAL4, UAS-mCherry* (BL 63148, Bloomington *Drosophila* Stock Center), *w; hsFLP, Act5C>y⁺>GAL4, UAS-GFP* (*Sekyrova et al., 2010*), *w; hsFLP, Act5C>y⁺>GAL4, UAS-GFP, UAS-Atf3^WT^* (*Donohoe et al., 2018*), *w; hsFLP, Act5C>y⁺>GAL4, UAS-myr-mRFP, UAS-Atf3^WT^* (recombined in this study), *w;; Ubi-GFP. E2f1^1-230^, Ubi-mRFP1.NLS.CycB^1-266^* (*Zielke et al., 2014*), *w;; BcF6:GFP* (*Tokusumi et al., 2009*), *w; Bc¹* (*Rizki et al., 1980*), *w; P[PTT-un1]vkgG454* (*vkg::GFP, Morin et al., 2001*), *w; UAS-Mp^RNAi^* (v38189, VDRC), *w; UAS-Mp^RNAi^ II* (v35431, VDRC), *w; UAS-Col4a1^RNAi^* (v28369, VDRC), *w, UAS-trol^RNAi^* (v22642, VDRC), *w; UAS-vkg^RNAi^* (v16986, VDRC), *w; UAS-LanA^RNAi^* (v18873, VDRC), *w; UAS-LanB1^RNAi^* (v23119, VDRC), *w; UAS-LanB2^RNAi^* (v42559, VDRC), *w; UAS-SPARC^RNAi^* (v16677, VDRC), *w; UAS-Mp^RNAi^, UAS-SPARC^RNAi^* (recombined in this study), *w; Hml:DsRed; a58-GAL4, UAS-GFP* (*Galko and Krasnow, 2004*, recombined in this study), *w; Hml:DsRed; mef2-GAL4, UAS-GFP* (*Ranganayakulu et al., 1996*, recombined in this study), *w, elav-GAL4; Hml:DsRed, UAS-GFP* (*Lin and Goodman, 1994*, recombined in this study), *eater¹* (*Bretscher et al., 2015*), *w; C7-GAL4, UAS-GFP, Hml:DsRed, UAS-atf3^WT^; eater¹* (recombined in this study), *w; HmlΔ-GAL4, UAS-GFP* (*Sinenko and Mathey-Prevot, 2004*), *w;; UAS-eater^RNAi^* (v4301, VDRC), *w;; UAS-Mp::GFP* (this study).

## Whole larval imaging

Third instar larvae were collected and washed in PBS and placed in glass dissection dishes filled with PBS on ice for 15 min for immobilization. Imaging was performed with an Olympus SZX-16 microscope fitted with a DP72 camera. GFP and RFP images were captured with the cellSens standard v1.11 software (Olympus, RRID:SCR_014551).

Stereotypical sessile tissue patterns were generated from larvae imaged under identical conditions. One larva was selected as a reference and every other image was aligned to match the reference in Adobe Photoshop CS5.5 (Adobe Systems, Inc, RRID:SCR_014199) using the Puppet warp tool. Aligned images were projected in Fiji v1.52i (*Schindelin et al., 2012*, RRID:SCR_003070) with 'Average intensity' projection, using the 'Fire' lookup table to enhance visualization.

For confocal imaging of whole fixed larvae, third instar wandering larvae were washed in 70% ethanol and were then injected with 4% paraformaldehyde with a sharpened glass injection capillary. The fixed larvae were immediately placed on a glass slide with a double-sided tape and imaged with an Olympus FV-1000 confocal microscope, with an UPlanSApo 10x (NA0.40) objective. Images were taken using the multi-area module of the FluoView FV-10ASW (Olympus, RRID:SCR_014215) software. Overlapping images were stitched in Fiji v1.52i and all single Z-planes were exported for both GFP (fat body) and DsRed (hemocytes) channels. Each individual confocal section was examined for GFP expression (fat body) and Hml:DsRed expression (hemocyte). The DsRed channel was locally recolored on each section when it was determined to overlap with the GFP signal to amber using Adobe Photoshop CS5.5, then all Z-planes were maximum projected. Sessile- and lymph gland hemocytes are shown in magenta.

## Tissue dissection and immunostaining

Fat body dissection from third instar larvae was performed in PBS by opening the posterior of the larvae and inverting the carcass. After removal of the intestine, larvae were fixed in 4% paraformaldehyde for 1.5 hr. Larval epidermal fillet samples were prepared according to *Brent et al., 2009*, fixed in 4% paraformaldehyde, followed by the gentle removal of most body wall muscles. Tissues were blocked in 0.5% BSA (A3059, Sigma-Aldrich) supplemented with 0.1% TritonX-100 (T8787, Sigma-Aldrich) in PBS. Wing disc dissection was performed as described in *Donohoe et al., 2018*. Primary antibody staining was performed overnight at 4°C on a nutating mixer with anti-α-Tubulin (mouse, 1:200, DSHB, AA4.3, RRID:AB_579593), anti-Hemese (mouse, 1:100, *Kurucz et al., 2003*), anti-L1 (mouse, 1:100, *Kurucz et al., 2007b*), anti-NimC1 (mouse, 1:100, *Kurucz et al., 2007a*), anti-Laminin (rabbit, 1:500, Abcam, ab47651, RRID:AB_880659), anti-phospho-histone H3 (rabbit, 1:500, Cell Signaling, Cat# 9701, RRID:AB_331535), anti-Endostatin (rat, 1:200, *Harpaz et al., 2013*) and

anti-cleaved-Dcp-1 (rabbit, 1:500, Cell Signaling, Cat# 9578, RRID:AB_2721060) antibodies diluted in the blocking solution. After washing, the samples were incubated with the corresponding Cy3- or Cy5-conjugated secondary antibodies (Jackson ImmunoResearch) for 1.5 hr at room temperature and counterstained with DAPI (1 μg/ml, 6335.1, Carl Roth GmbH) to visualize nuclei. The fat bodies were dissected from the carcass and mounted in Dabco-Mowiol (Sigma-Aldrich). Tissues were imaged on an Olympus FV-1000 confocal microscope with UPlanSApo 10x (NA0.40), UPlanSApo 20x (NA0.75), UPlanSApo-O 20x (NA0.85), UPlanFLN-O 40x (NA1.30) and UPlanSApo-O 60x (NA1.35) objectives using the FluoView FV-10ASW software. All fat body images represent Z-projections (unless otherwise indicated), which were generated with Fiji v1.52i.

For the pseudocoloring of hemocyte and adipocyte nuclei, confocal Z-stacks of the DAPI channel were divided into surface sections (~0–8 μm from the surface) and deeper sections (>8 μm from the surface), which were projected and exported separately using Fiji v1.52i. The surface projection containing the hemocyte nuclei was then colored amber, while the deeper projection with the adipocyte nuclei was colored magenta. The two projections were merged in Adobe Photoshop CS5.5.

### *Leptopilina boulardi* infection

Flies were reared on the following diet: 8% cornmeal, 1% agar, 4% yeast, 5% saccharose and 0.16% methylparaben. Fifty early third instar larvae were placed in an infection chamber with 100 female *L. boulardi* wasps for 15 min (Bajgar et al., 2015). After the removal of the wasps, the larvae were kept at 25°C for 24 or 48 hr before bleeding or dissection.

### Hemocyte isolation and staining

Third instar larvae were bled into 1x PBS containing 0.01% n-phenylthiourea (31056, Sigma-Aldrich) on 12 spot glass slides (HM-101, Hendley-Essex). Allowing hemocytes to adhere for 45 min, samples were then fixed in acetone. Blocking was performed with 0.5% BSA in 1x PBS for 30 min. Hemocytes were stained with anti-Hemese (mouse, 1:100, Kurucz et al., 2003) or anti-L1 (mouse, 1:100, Kurucz et al., 2007b) antibodies and anti-mouse-Cy3 conjugated secondary antibodies (1:500, Jackson Immunoresearch). Nuclei were counterstained with DAPI. Samples were mounted on glass slides in Dabco-Mowiol 4–88 and imaged on an Olympus FV-1000 confocal microscope with UPlanSApo 10x (NA0.40) and UPlanSApo-O 20x (NA0.85) objectives.

### Transmission electron microscopy

Fat bodies were fixed in 2.5% glutaraldehyde diluted in 100 mM phosphate buffer (PB), washed in 100 mM PB and postfixed in 2% osmium tetroxide in PB for 1 hr on ice. Contrasting was performed with 2% uranyl acetate, after which the samples were dehydrated in ethanol and embedded in acetone-resolved araldite. Electron microscopy was performed with an EM 109 (Zeiss) microscope.

### Cloning of Eater and Mp expression plasmids and generation of UAS-Mp::GFP *Drosophila* line

To express C-terminally GFP- or FLAG-tagged Eater and Multiplexin proteins under UAS control, *eater* and *Mp* cDNA was cloned into pENTR4-dual vector between *BamH*I and *Not*I sites, and subsequently recombined using LR Clonase II (11791–020, Life Technologies) into pTWG (Mp and Eater) and pTWF (Mp) vectors, respectively (T. Murphy, *Drosophila* Genomic Resource Center). The *UAS-Mp::GFP* transgenic flies were obtained by standard P-element-mediated germline transformation of pTWG-Mp plasmid into *w[1118]* *Drosophila* embryos (Genetics Fly Facility, The University of Cambridge, UK). Multiple transformants were recovered and tested, all showing comparable Mp::GFP expression.

### S2 cell culture and cell lysis

Schneider 2 (S2) cells (*Drosophila* Genomic Resource Center, RRID:CVCL_Z992) were cultured at 25°C in Shields and Sang M3 insect medium (S8398-1L, Sigma-Aldrich) containing 8% fetal bovine serum (Gibco, Life Technologies) without antibiotics. There are no verified reports of Mycoplasma infection in S2 cells (Cherbas and Gong, 2014). S2 cells were only used to express transgenic proteins for biochemical experiments. All functional data were obtained from in vivo studies in a *Drosophila melanogaster* model. Cells were transfected using TransIT Insect transfection reagent (MIR

6100, Mirus). Expression of UAS-driven cDNAs was induced by co-transfection with a pWA-GAL4 plasmid expressing GAL4 under an *actin5C* promoter. Cells were lysed 36 hr after transfection in lysis buffer containing 50 mM Tris-HCl (pH 7.8), 150 mM NaCl, 1 mM EDTA (pH 8.0), 1% Triton X-100, 0.01% Igepal, and protease inhibitors (Roche Applied Science). Protein concentration was quantified using Bradford reagent (K015.1, Roth GmBH) according to manufacturer's instructions.

## Co-immunoprecipitation and immunoblotting

For each sample, 1 mg of S2 cell protein lysate was incubated with GFP-Trap beads (gtma-20, Chromotek, RRID:AB_2631406) overnight. Following five washes with Lysis buffer, bound proteins were eluted with Glycine-HCl on 37°C, followed by neutralization. Proteins resolved on 10% SDS-PAGE were detected by immunoblotting with anti-Flag M2 (mouse, 1:1000, F1804, Sigma Aldrich, RRID: AB_262044), anti-GFP (rabbit, 1:5000, TP401, Acris, RRID:AB_2313770) and anti-α-Tubulin (mouse, 1:1000, DSHB, AA4.3, RRID:AB_579593) antibodies, followed by incubation with corresponding HRP-conjugated secondary antibodies (Jackson Immuno Research). Chemiluminescent signal was captured using ImageQuant LAS4000 reader (GE Healthcare, RRID:SCR_014246).

Hemolymph was collected from twenty third instar larvae per replicate in 100 µL PBS. After centrifugation, 5000 rpm for 10 min, the supernatant was precipitated with 100 µL of ice cold acetone for 1 hr at −20°C. Afterwards, the proteins were pelleted and resuspended in 50 µL in Lysis buffer (see above), and equal volumes were resolved on 10% SDS-PAGE. Immunoblotting was carried out with anti-Endostatin (rat, 1:1000, *Harpaz et al., 2013*) and anti-PPO1 (rabbit, 1:750, *Jiang et al., 1997*) antibodies.

## Cell quantification and statistical analysis

Fat body associated hemocytes were quantified based on Z-projection images taken with Olympus FV-1000 confocal microscope fitted with UPlanSApo 20x (NA0.75) objective. The area imaged for comparisons is depicted in *Figure 1—figure supplement 1B*. The area selected is devoid of hemocytes in wild-type larvae, and lacks non-adipose tissue, such as the gonad precursors or the tracheal branches found at the posterior end. Quantification was performed manually using the CellCounter plugin in Fiji v1.52. For quantification of circulating hemocytes images were acquired using the UPlanSApo 10x (NA0.40) objective. One representative image per larva was processed for cell counting. DAPI positive nuclei were counted with CellProfiler (*Kamentsky et al., 2011*, RRID:SCR_007358), using a customized pipeline. Lamellocyte percentage was determined by comparing the number of L1-positive cells to all counted DAPI-positive nuclei.

For the quantification of sessile hemocytes, whole larval images aligned to a reference (see 'Whole larval imaging' section) were analyzed. These images were loaded in Fiji v1.52i as hyperstacks, containing both control larvae and the genotypes of interest. Using the Rectangle tool, an area was selected where no hemocytes were present, where the average pixel intensity was measured for each control larva and averaged across larvae to establish background levels. Using the Rectangle tool, selections corresponding to single sessile stripes in the A4, A5, A6 and A7 segments were made on control larvae, and the average intensity of the Hml:DsRed signal was measured for the relevant genotypes within this selection. From every measurement the previously established background values were subtracted, then for each larva the four corrected intensity values were added together.

Statistical analysis and plotting were carried out with Prism 6 (GraphPad, RRID:SCR_002798), using one-way ANOVA assuming unequal variances with multiple comparisons or unpaired Student's t-test. For comparison of cell counts, one-way Kruskal-Wallis test with Dunn's multiple comparison or nonparametric two-tailed Mann-Whitney test were used. For fat-body-associated hemocytes, every data point represents the number of attached hemocytes within the quantified area, while for circulating hemocytes and sessile hemocyte quantifications, values were normalized to control mean (represented as 1) and shown on the graphs as fold change (F.C.). All experiments represent at least two independent temporal replicates.

Sample-size criteria was estimated *post hoc* with G*Power 3.1 (*Faul et al., 2009*). All significantly different datasets exceeded 0.99 Power (1-beta) with respective sample sizes, means and standard deviations.

## Acknowledgements

We thank Talila Volk, Michael R Kanost, Bruno Lemaitre, István Andó, the Bloomington *Drosophila* Stock Center supported by NIH grant NIH P40OD018537 (BDSC, Bloomington, IN, USA), the *Drosophila* Genomics Resource Center supported by NIH grant 2P40OD010949 (DGRC, Bloomington, IN, USA), the Vienna *Drosophila* Resource Center (VDRC, Vienna, Austria), and the Developmental Studies Hybridoma Bank (DSHB, Iowa City, IA, USA) for fly stocks, plasmids, cell line, and antibodies. We are grateful to Tomáš Doležal and Pavla Nedbalová for access to and sharing experience with the parasitoid wasp infection model at the University of South Bohemia (Ceske Budejovice, Czech Republic). We also thank Marek Jindra (Biology Center CAS, Czech Republic) and the Bioscience Imaging and Histology Unit of the Institute of Entomology (Biology Center CAS, Czech Republic) for microscope access. We are grateful to Steffen Erkelenz for advice on cloning and immunoprecipitation experiments, Nils Teuscher and Tina Bresser for excellent technical assistance, and the entire Uhlirova laboratory for discussion. This work was funded by UH 243/3–1 project to M.U and under Germany's Excellence Strategy – CECAD, EXC 2030 – 390661388 both from the Deutsche Forschungsgemeinschaft (DFG, German Research Foundation).

## Additional information

### Funding

| Funder | Grant reference number | Author |
|---|---|---|
| Deutsche Forschungsgemeinschaft | UH 243/3-1 | Mirka Uhlirova |
| Deutsche Forschungsgemeinschaft | EXC 2030 - 390661388 | Mirka Uhlirova |

The funders had no role in study design, data collection and interpretation, or the decision to submit the work for publication.

### Author contributions

Gábor Csordás, Conceptualization, Resources, Data curation, Validation, Investigation, Visualization, Methodology, Writing - original draft, Writing - review and editing; Ferdinand Grawe, Methodology; Mirka Uhlirova, Conceptualization, Resources, Supervision, Funding acquisition, Methodology, Writing - original draft, Project administration, Writing - review and editing

### Author ORCIDs

Gábor Csordás (iD) https://orcid.org/0000-0001-6871-6839
Mirka Uhlirova (iD) https://orcid.org/0000-0002-5735-8287

### Decision letter and Author response

Decision letter https://doi.org/10.7554/eLife.57297.sa1
Author response https://doi.org/10.7554/eLife.57297.sa2

## Additional files

### Supplementary files

- Transparent reporting form

### Data availability

All data generated or analyzed during this study are included in the manuscript and supporting files. Source Data files contain raw data for all Figures where relevant.

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
