## [Decision Letter]

**Acceptance summary:**

*Drosophila* hemocytes come in and out of hematopoietic microenvironments within which they proliferate or differentiate. Despite its importance in hematopoiesis, the molecular mechanism underlying hemocyte-microenvironment interaction remains poorly understood. Here, Uhlirova and colleagues identify critical roles for basement membrane proteins, eater and Multiplexin, in the formation of hematopoietic compartments, providing fundamental insights into how hematopoietic niches arise and maintain.

**Decision letter after peer review:**

Thank you for submitting your article "Eater cooperates with Multiplexin to drive the formation of hematopoietic compartments" for consideration by *eLife*. Your article has been reviewed by three peer reviewers, one of whom is a member of our Board of Reviewing Editors, and the evaluation has been overseen by Utpal Banerjee as the Senior Editor. The following individuals involved in review of your submission have agreed to reveal their identity: Dan Hultmark (Reviewer #3).

The reviewers have discussed the reviews with one another and the Reviewing Editor has drafted this decision to help you prepare a revised submission.

Summary:

In this manuscript, Uhlirova and colleagues have reported an interesting molecular mechanism underlying the hemocyte recruitment to its environments, which bears a clinical potential to human immune cell biology. The authors first focused on a novel phenomenon that overexpression of Atf3 recruits hemocytes to the fat body while altering patterns of hematopoietic pockets in segmentally repeated epidermal sites. The authors went on to understand a molecular basis for the redirection of hemocytes and discovered that Collagen XV/XVIII type protein Multiplexin (Mp) is required for the Atf3-mediated hemocyte homing dependent on the Eater expression in hemocytes. Given that overexpression of Mp in the fat body disrupts the overall pattern of hematopoietic pockets and that Mp is detected in the hemocyte-free hemolymph, the authors raised a valid hypothesis that Mp can be deposited to the hemolymph, which consequently promotes the systemic hemocyte homing to the sessile pockets.

Essential revisions:

As you will find below, all of the reviewers found the study interesting and agreed on the significance of your work. At the same time, a number of critical criticisms were raised, which require additional data analyses and new experiments. The full comments of the reviewers are attached to provide further details.

1) Identify division and differentiation rates of hemocytes in the Atf3 or Mp expression background.

- As indicated in the comments below (reviewer 1 and reviewer 2), it will be critical to show the redirection of hemocytes to a new tissue would recapitulate hematopoiesis in the hematopoietic pocket.

2) Improve figure images and provide details in figure legends.

- There are multiple concerns about this issue from three reviewers. Please find the comments below for details.

- Of note, it will be important to adequately address reviewer 3's Essential revisions 3 and 4 where the reviewer raised concerns about images showing the fat body associated hemocytes.

3) Provide quantitation of results.

- All the reviewers asked for proper quantitations of data for clarity. Please find the details below.

Reviewer #1:

In this manuscript, Uhlirova and colleagues have reported a critical role for basement membrane proteins in the recruitment of hemocytes in the fat body. The authors first focused on a novel phenomenon that overexpression of Atf3 recruits hemocytes to the fat body while altering patterns of hematopoietic pockets in segmentally repeated epidermal sites. Interestingly, hemocytes attached to the fat body undergo mitosis similar to the ones found in hematopoietic pockets and exhibit normal morphologies. Moreover, hemocytes associated with the fat body relocate to the circulation upon wasp infestation, indicating that these hemocytes display a normal hematopoietic program reminiscent of sessile hemocytes. The authors further went on to understand a molecular basis for the attachment and discovered that Collagen XV/XVIII type protein Multiplexin (Mp) is required for the Atf3-mediated hemocyte homing which is dependent on the Eater expression in hemocytes. Supporting these findings, the authors validated a physical interaction between Mp and eater by biochemical experiments. Given that overexpression of Mp in the fat body disrupts the overall pattern of hematopoietic pockets and that Mp is detected in the hemocyte-free hemolymph, the authors raised a valid hypothesis that Mp can be deposited to the hemolymph critical for the systemic hemocyte homing to the sessile pockets.

This study provides a fundamental insight into molecular mechanisms underlying the hematopoietic environment essential for immune cell development and its function and prompts future studies linking the ECM landscape with the hematopoietic pocket formation and hemocyte development.

Essential revisions:

1) The precise quantitation of data and presenting proper controls will enhance the clarity of the manuscript.

1.1) Figure 1C-D': In Figure 1E, the authors quantified FBAH in imaged fat bodies. However, the number of fat body cells per one image could be variable depending on the size of the fat body, dissecting methods or imaging and so on. For example, the mean value shown in Figure 1E is different from Figure 2I even though these data represent the same genotype. It will be important to deliberate on the precise quantitation method for the FBAH phenotype as it is one of the most important findings described in the manuscript.

1.2) The control shown in Figure 1E is identical to the one shown in Figure 2I and Figure 3—figure supplement 2E. It will be essential to ALWAYS repeat relative controls side by side with associating experiments.

1.3) Figure 2J-K', the number of lamellocytes and the percentage of melanotic capsule formation need to be quantified.

1.4) In the same vein, Figure 3—figure supplement 1C-K' requires proper quantitation with enough biological replicates (n) to draw a conclusion.

1.5) Figure 1A-B, Figure 3G-H, Figure 4A,G,H, Figure 4—figure supplement 1A-F, Figure 5C-D: authors have shown the relocation of sessile hemocytes. Again, it is one of the main phenotypes shown in the paper; however, none of these are quantitated. The patterns of sessile hemocytes are quite variable even amongst controls. Therefore, authors need to quantify the phenotype with their own measure and with enough sample size.

1.6) In Figure 4D-E': though it has been shown in the previous study that eater is essential for the hemocyte homing and that eater is primarily expressed in hemocytes, C7-gal4, hml-gal4 (dual gal4) is a different genetic background and can cross-react. Therefore, additional controls (C7-gal4, hml-gal4/+, C7-gal4, hml-gal4; eater RNAi) are required at least to be represented as quantitation data.

2) It is not clear whether the overall hemocyte number (or hemocyte development) is changed upon Atf3_WT or Mp::GFP expression, which may, in turn, modify the circulating or sessile hemocyte population.

2.1) Figure 2C: To claim whether the epidermal sites are more supportive of the proliferation of hemocytes than those in the fat body, the ratio of mitotic cells needs to be quantified as shown in Makhijani et al., 2017. And without showing the number of total hemocytes, it is hard to understand which site is more prominent in the proliferation and differentiation of hemocytes.

2.2) The total number of hemocytes together with circulating hemocytes needs to be shown in Atf3_wt or Mp::GFP expression (Figure 1—figure supplement 1C, Figure 5E). Moreover, the number of circulating and total hemocytes in Atf3_WT; MpRNAi rescue background should be indicated.

3) In Figure 2E-E', it is not clear whether crystal cells express high levels of NimC1 in the images as NimC1-positive membranes are anyway juxtaposed. It will be great to have a bleeding of FABH or high mag of crystal cells with better resolution.

4) Though authors have shown that FBAHs are relocated to the circulation upon wasp parasitism, these data are not sufficient to conclude that the fat body FBAH follows the identical hematopoietic program to sessile blood cells. It is possible that FBAHs relocation is less frequent than regular sessile hemocytes found in the epidermis/neuron due to the tight association. It will be important to show whether FBAHs can be reattached after physical disturbances as shown in Makhijani et al., 2011.

Reviewer #2:

This very well written and interesting manuscript addresses the clinically relevant question of what molecules drive immune cells' interaction with environments that promote their proliferation and differentiation. The authors use the basic system of *Drosophila* and provide convincing evidence that an ECM component, Multiplexin, is necessary for the recruitment of *Drosophila* immune cells (hemocytes) to the previously identified hematopoietic pockets and for their recruitment to a new position on the fat body where they are can divide and differentiate as they do in the native environment. They define the phagocytosis receptor Eater, which has previously been found to be crucial for hemocyte localization to the hematopoietic pockets, as the partner in mediating hemocyte binding to Multiplexin in both locations. I think this work is exciting and mostly well conducted and deserves to be published in *ELife*, with some alteration of the text to reflect that some of the current conclusions require a modest amount of new data and analysis.

1) The paper in places implies that Multiplexin is capable of redirecting hemocytes to a new location in which they can function as in the hematopoietic pocket. This finding would be extremely exciting if true. But though the authors show relocalization to the fat body in the experiment in which they induce clones of Mp (Figure 5G-H), they do not assess the division and differentiation capacity of the hemocytes there. Thus Atf3 expression may induce other changes in the adipocytes than just Mp expression to allow Fat Body associated hemocytes (FBAH) to divide and differentiate there (as seen in Figure 2C-F); Mp may thus be required for hemocyte adhesion yet not sufficient to induce all niche dependent functions. To address sufficiency for proliferation the authors could do a pH3 staining upon the induction of clones in strains they already have (those used in Figure 5G-H). To address sufficiency for differentiation, they could put one copy of Bc1 into that background.

Alternatively they should alter the paper to make clearer that they have not explicitly investigated if or shown that the ECM component, Mp, drives hematopoiesis in the Title, the Abstract, the Introduction, the Discussion section (specifically "indistinguishable").

2) They conduct experiments to assess the importance of Mp for attachment to the endogenous sessile niches. Their conclusion (subsection “Similar mechanisms drive sessile hematopoietic pocket formation and FBAH adhesion”) that the knockdown of Mp by RNAi produces a phenotype strikingly resembling that seen in the eater1 mutant (and thus that Mp is required for all attachment to endogenous hematopoetic environments) doesn't seem supported by the comparison of Figure panels 4A' to 4H'. While the dorsal patches seem to be gone in the Mp knockdown, it appears as if there is still a repeated hemocyte pattern on the lateral sides in the larvae shown. Having a close up of the lateral and dorsal niches for these genotypes instead of the GFP channel (whose relevance as a marker of the driver pattern is also not explained in the figure legends) would clarify this issue. It is clear there is a strongly reduced attachment in the absence of Mp, but if this the same in all the different endogenous hematopoetic locations is important to determine. Given that Mp has been reported to be expressed in the heart tube, next to where the dorsal patches are, and not to my knowledge near the PNS, which flanks the lateral ones, would fit with there being a variable effect in the different regions. This Figure has no quantitation, which also weakens this conclusion. If they do not have the data in hand to make the new figure/ do the quantitation, then they should soften the conclusion that this is the same as the eater1 phenotype and address textually that the effect on the lateral patches has not been rigorously assessed.

Reviewer #3:

I believe this is a great paper, with exciting science, but I find it unnecessarily difficult to read and to assess critically. I was easily distracted while reading it and I may have missed important points.

Essential revisions:

1) Conceptually, it would be helpful to give the reasons for choosing to overexpress Atf3 in the fat body. I would also like to see more background information about this transcription factor in the Introduction.

2) I also miss a discussion about the possible physiological role of the system described here. Is Atf3 also involved in defining hemocyte docking sites elsewhere? Or should I regard Atf3 overexpression as an artificial but useful experimental system for the study of hematopoiesis?

3) In general, the figure legends give little or no information about the experiments shown. Instead they merely describe what conclusions I am expected draw, something that is anyway already explained in the text. There are many examples:

a) In Figure 1, the reader has to guess how the different colors are generated. I assume that the green color comes from a UAS-GFP construct, driven by the C7 fat body-specific driver. An Hml:DsRed construct is apparently also involved. It should give red fluorescence to plasmatocytes. Apparently, that this has been converted to the "magenta" color, here labeled "Hemocyte" (I comment the "amber"-colored hemocytes below). Similar difficulties complicate interpretations in other figures too.

b) The units on y-axes are sometimes unclear. Does "Number of FBAHs" (Figure 1, Figure 2, and Figure 3) refer to the number per unit area as defined in Figure 1—figure supplement 1? And what is "Circulating hemocytes (F.C.)" in Figure 1, Figure 1—figure supplement 1 and Figure 5?

c) If "Hemese" (in Figure 2 and Figure 5) refers to antibody staining, for general visualization of hemocytes, then say so.

d) What is "myr-RFP" (Figure 3A and B), and what does it signify?

I will stop giving examples here, but my progress through the manuscript was slowed down dramatically by the fact that I constantly had to make guesses about what I could see in the figures.

4) How was the "amber"-coloring of hemocytes generated in Figure 1B (and elsewhere)? The text in the Materials and methods section seems to indicate that these cells were simply enhanced in this way via Photoshopping. That does give a striking effect, but is not valid evidence that these cells are attached to the fat body.

5) On a similar note, the intracellular vacuoles in Figure 3F are colored "cyan", presumably by Photoshop, to generate the impression that they are contiguous with the similarly colored "pericellular" space under the basement membrane. This is a potentially dangerous way to fool the eye. Do the authors have evidence that these systems are contiguous, or that they are related in other ways? Is there evidence that the associated electron-dense material consists of extracellular matrix proteins?

6) As shown here, several extracellular matrix proteins are enriched over Atf3-expressing cells. Does that mean that the basement membrane is thickened? Zooming in on Figure 3F I get an impression that this may be the case.

7) I find no information about how the Atf3 clones were generated for Figure 1F, and very rudimentary information about the clones in Figure 3 and Figure 5.

8) The magnification and resolution is not sufficient to show the filopodia and lamellipodia in Figure 2A, 2B, Figure 5G and 5H, even when I print out these figures over entire pages. They are even difficult to see when I zoom them in on the computer.

9) Has the C7 driver ever been described? The authors refer to a previous article from the same lab (Rynes et al., 2012), where I only find a reference back to a paper by Grönke et al., (2003). However, C7 is not mentioned there. How specific is the C7 driver, and why is it used?

[Editors' note: further revisions were suggested prior to acceptance, as described below.]

Thank you for resubmitting your work entitled "Eater cooperates with Multiplexin to drive the formation of hematopoietic compartments" for further consideration by *eLife*. Your revised article has been evaluated by Utpal Banerjee (Senior Editor) and a Reviewing Editor.

The manuscript has been improved but there are some remaining issues that need to be addressed before acceptance, as outlined below:

Your revised manuscript was evaluated by original reviewers (reviewer #1 and reviewer #2) and an additional reviewer (reviewer #4). All the reviewers agreed that now the manuscript is greatly strengthened and is ready for publication after incorporating a few minor changes as suggested by reviewer #2 and #4. Reviewer #4 recommended providing additional statistical information in Materials and methods section and applying generalized linear modeling for count data.

Reviewer #1:

With an extensive revision, the authors significantly improved the manuscript and satisfactorily addressed all my concerns. Additional data and quantitation looked convincing, and detailed descriptions of the Materials and methods section and Figure legends enhanced its readability.

Reviewer #2:

The authors have fully addressed my major concerns. I believe the work is greatly strengthened and now makes a convincing case that Mp is capable of inducing the hemocyte proliferation and differentiation in the new fat body location. The quantitation of the Mp knockdown showing the effect on sessile hemocytes has also been conducted, fully anchoring their conclusions on Mp importance for hemocyte localization at endogenous sites of hematopoiesis.

I look forward to seeing this exciting work online soon.

Reviewer #4:

This manuscript provides novel insight into the interaction of *Drosophila* blood cells, the hemocytes, with other tissues during the formation of sessile hemocytes clusters. The manuscript provides evidence on a key role of Multiplexin in the formation of these clusters, which, as this manuscript and previous work shows, are important sites for hemocyte proliferation and probably also differentiation. The reviewers in the first round of assessment have pointed out valid concerns regarding the manuscript and, in my opinion, the authors have done a good job addressing these concerns. I have only few remaining points/questions for the authors.

On some occasions, the number of animals tested seem quite low, especially when *Drosophila* material is not exactly scarce. However, I don't see this as a decisive problem in this study, one reason being that differences between controls and experimental crosses are in general quite clear. Also, multiple experiments conducted back each other up. However, could authors provide more detailed information on how experiments were replicated? Specifically, were the animals studied from the progeny of one cross/the same patch of crosses or were experiments replicated temporally? This could be clarified in the Materials and methods section since it helps in assessing the reproducibility of the results.

---

## [Author Response]

Reviewer #1:In this manuscript, Uhlirova and colleagues have reported a critical role for basement membrane proteins in the recruitment of hemocytes in the fat body. The authors first focused on a novel phenomenon that overexpression of Atf3 recruits hemocytes to the fat body while altering patterns of hematopoietic pockets in segmentally repeated epidermal sites. Interestingly, hemocytes attached to the fat body undergo mitosis similar to the ones found in hematopoietic pockets and exhibit normal morphologies. Moreover, hemocytes associated with the fat body relocate to the circulation upon wasp infestation, indicating that these hemocytes display a normal hematopoietic program reminiscent of sessile hemocytes. The authors further went on to understand a molecular basis for the attachment and discovered that Collagen XV/XVIII type protein Multiplexin (Mp) is required for the Atf3-mediated hemocyte homing which is dependent on the Eater expression in hemocytes. Supporting these findings, the authors validated a physical interaction between Mp and eater by biochemical experiments. Given that overexpression of Mp in the fat body disrupts the overall pattern of hematopoietic pockets and that Mp is detected in the hemocyte-free hemolymph, the authors raised a valid hypothesis that Mp can be deposited to the hemolymph critical for the systemic hemocyte homing to the sessile pockets.This study provides a fundamental insight into molecular mechanisms underlying the hematopoietic environment essential for immune cell development and its function and prompts future studies linking the ECM landscape with the hematopoietic pocket formation and hemocyte development.Essential revisions:1) The precise quantitation of data and presenting proper controls will enhance the clarity of the manuscript.1.1) Figure 1C-D': In Figure 1E, the authors quantified FBAH in imaged fat bodies. However, the number of fat body cells per one image could be variable depending on the size of the fat body, dissecting methods or imaging and so on. For example, the mean value shown in Figure 1E is different from Figure 2I even though these data represent the same genotype. It will be important to deliberate on the precise quantitation method for the FBAH phenotype as it is one of the most important findings described in the manuscript.

We expanded the Materials and methods section to accurately describe how the quantifications were performed. The area of the fat body assessed throughout the study and across the different genotypes was specifically selected because of its flat surface and absence of additional tissues that can associate with this organ (such as salivary gland in the anterior part or the gonads or tracheae at the posterior end). Furthermore, while structural variances could indeed be observed in response to manipulating various genes, such as *SPARC* or *vkg*, the overall size of the adipose tissue was not noticeably affected by co-expression of additional transgenes with *UAS-Atf3WT*.

Regarding the differences in FBAHs numbers in Figure 1E and Figure 2I, we consider different food composition as an important and likely source of variability in the number of FBAHs between the two datasets mentioned. The wasp infection experiments were performed in the lab of Dr. Tomas Dolezal (University of South Bohemia, Czech Republic). This means that flies/larvae were reared on different fly food than the larvae throughout the manuscript. The numbers therefore should be interpreted within the context of the particular experiment. We now describe in detail the experimental setup for the wasp infestation experiments in the Materials and methods section including the fly food composition.

1.2) The control shown in Figure 1E is identical to the one shown in Figure 2I and Figure 3—figure supplement 2E. It will be essential to ALWAYS repeat relative controls side by side with associating experiments.

The control numbers (indicated in gray) were repeated because these genotypes were dissected and quantified as parts of the same experiments to allow comparison across genotypes. To align with the order of the data presentation, the data had to be “artificially” split. We now provide additional data with separate controls for each experiment shown. In the case of Figure 3K and Figure 3—figure supplement 1I we indicate in the supporting datasets that the results belong to the same experiment.

1.3) Figure 2J-K', the number of lamellocytes and the percentage of melanotic capsule formation need to be quantified.

We now include the quantification of lamellocytes (Figure 2J). The capsule formation rate cannot be assessed as the wasp infestation experiments were performed in the lab of Dr. Tomas Dolezal (University of South Bohemia, Czech Republic) that is currently not accessible to us due to the travel ban for the University of Cologne employees due to COVID-19 pandemic. We would like to note that the data shown in Figure 2J-O was only included to indicate that lamellocyte differentiation and encapsulation is not noticeably affected in *C7>Atf3WT* larvae, and we did not intend to draw conclusions about the efficiency of the immune response. We adjusted the wording of the manuscript to indicate this.

1.4) In the same vein, Figure 3—figure supplement 1C-K' requires proper quantitation with enough biological replicates (n) to draw a conclusion.

We performed the quantifications which are now included in Figure 3—figure supplement 1I and reflect on the data in the text.

1.5) Figure 1A-B, Figure 3G-H, Figure 4A,G,H, Figure 4—figure supplement 1A-F, Figure 5C-D: authors have shown the relocation of sessile hemocytes. Again, it is one of the main phenotypes shown in the paper; however, none of these are quantitated. The patterns of sessile hemocytes are quite variable even amongst controls. Therefore, authors need to quantify the phenotype with their own measure and with enough sample size.

We devised a quantification method of the sessile hemocyte patterns based on image analysis. The description of the quantification method is now included in the Materials and methods section. The results are included in the relevant figures. For data shown in Figure 4—figure supplement 1A-F, we adapted relevant statements in the Results section.

1.6) In Figure 4D-E': though it has been shown in the previous study that eater is essential for the hemocyte homing and that eater is primarily expressed in hemocytes, C7-gal4, hml-gal4 (dual gal4) is a different genetic background and can cross-react. Therefore, additional controls (C7-gal4, hml-gal4/+, C7-gal4, hml-gal4; eater RNAi) are required at least to be represented as quantitation data.

We performed quantifications comparing *C7>Hml>Atf3WT* and *C7>Hml>Atf3WTeaterRNAi* and included the results in the manuscript and in Figure 4H-J.

2) It is not clear whether the overall hemocyte number (or hemocyte development) is changed upon Atf3_WT or Mp::GFP expression, which may, in turn, modify the circulating or sessile hemocyte population.2.1) Figure 2C: To claim whether the epidermal sites are more supportive of the proliferation of hemocytes than those in the fat body, the ratio of mitotic cells needs to be quantified as shown in Makhijani et al., 2017. And without showing the number of total hemocytes, it is hard to understand which site is more prominent in the proliferation and differentiation of hemocytes.

The manuscript does not state that “the epidermal sites are more supportive of the proliferation of hemocytes than those in the fat body”, only that the epidermally associated hemocytes proliferate more than freely circulating ones, citing the data and conclusions from Makhijani et al., 2017. Our intention was to determine the behavior of hemocytes on the fat body surface, and the phenomena of hemocyte proliferation and plasmatocyte-crystal cell trans-differentiation are indicative of an unchallenged, homeostatic hemocyte developmental program.

Furthermore, we indicate in the Discussion section that differences likely exist between FBAHs and sessile hemocytes, but the mechanistic basis of their attachment is conserved.

2.2) The total number of hemocytes together with circulating hemocytes needs to be shown in Atf3_wt or Mp::GFP expression (Figure 1—figure supplement 1C, Figure 5E). Moreover, the number of circulating and total hemocytes in Atf3_WT; MpRNAi rescue background should be indicated.

The experiments combining *Atf3WT* and *MpRNAi* expression in the fat body were aimed to show that on one hand the hemocyte attachment to the fat body is dependent on Mp, and on the other hand, when hemocytes cannot attach to the adipose tissue the pattern of the sessile hematopoietic compartment is restored, for which we now include additional data in Figure 3L-O. We would also like to note that the comparison of total hemocyte numbers using the methods from Petraki et al., 2015 may not reflect the accurate values, as those are optimized for circulating and sessile cells, and not hemocytes attached to the internal organs.

3) In Figure 2E-E', it is not clear whether crystal cells express high levels of NimC1 in the images as NimC1-positive membranes are anyway juxtaposed. It will be great to have a bleeding of FABH or high mag of crystal cells with better resolution.

We exchanged the panels in Figure 2E-E’ to better show the colocalization of the Bc:GFP signal and the NimC1 staining on the intermediate hemocytes.

4) Though authors have shown that FBAHs are relocated to the circulation upon wasp parasitism, these data are not sufficient to conclude that the fat body FBAH follows the identical hematopoietic program to sessile blood cells. It is possible that FBAHs relocation is less frequent than regular sessile hemocytes found in the epidermis/neuron due to the tight association. It will be important to show whether FBAHs can be reattached after physical disturbances as shown in Makhijani et al., 2011.

We did not intend to suggest that FBAHs and sessile hemocytes behave “identically”, rather that tissue/fat bodyattached hemocytes seem to behave in a similar fashion. To reflect this more accurately, we changed the wording of the manuscript. On a second note, similarly to Essential revisions 2.2, the physical mobilization assay described in Makhijani et al., 2011 was specifically created to release sessile hemocytes which adhere to the epidermis and are sensitive to external force (e.g. brush strokes, vortexing). FBAHs on the other hand are on the surface of the adipose tissue, which is floating in the hemolymph, and are therefore much less prone to these insults.

Reviewer #2:This very well written and interesting manuscript addresses the clinically relevant question of what molecules drive immune cells' interaction with environments that promote their proliferation and differentiation. The authors use the basic system of *Drosophila* and provide convincing evidence that an ECM component, Multiplexin, is necessary for the recruitment of *Drosophila* immune cells (hemocytes) to the previously identified hematopoietic pockets and for their recruitment to a new position on the fat body where they are can divide and differentiate as they do in the native environment. They define the phagocytosis receptor Eater, which has previously been found to be crucial for hemocyte localization to the hematopoietic pockets, as the partner in mediating hemocyte binding to Multiplexin in both locations. I think this work is exciting and mostly well conducted and deserves to be published in ELife, with some alteration of the text to reflect that some of the current conclusions require a modest amount of new data and analysis.1) The paper in places implies that Multiplexin is capable of redirecting hemocytes to a new location in which they can function as in the hematopoietic pocket. This finding would be extremely exciting if true. But though the authors show relocalization to the fat body in the experiment in which they induce clones of Mp (Figure 5G-H), they do not assess the division and differentiation capacity of the hemocytes there. Thus Atf3 expression may induce other changes in the adipocytes than just Mp expression to allow Fat Body associated hemocytes (FBAH) to divide and differentiate there (as seen in Figure 2C-F); Mp may thus be required for hemocyte adhesion yet not sufficient to induce all niche dependent functions. To address sufficiency for proliferation the authors could do a pH3 staining upon the induction of clones in strains they already have (those used in Figure 5G-H). To address sufficiency for differentiation, they could put one copy of Bc1 into that background.Alternatively they should alter the paper to make clearer that they have not explicitly investigated if or shown that the ECM component, Mp, drives hematopoiesis in the Title, the Abstract, the Introduction, the Discussion section (specifically "indistinguishable").

We now include experiments demonstrating the cell proliferation as well the presence of crystal cells among hemocytes attached to Mp::GFP overexpressing fat body clones (Figure 6C-E). We would like to add that while hemocyte division does take place in the clusters attached to the Mp::GFP clonal cells, the amount of attached hemocytes and the size of the clones vary among experimental replicates making the quantification of these phenomena problematic.

Furthermore, we include new data showing that Mp::GFP overexpression in the wing imaginal disc, also causes the specific attachment of hemocytes, further strengthening the evidence for the crucial role of Mp to facilitate hemocyte-tissue association (Figure 6F-G). Additionally, we reworded the cited conclusions to more appropriately describe the phenotypes observed.

2) They conduct experiments to assess the importance of Mp for attachment to the endogenous sessile niches. Their conclusion (subsection “Similar mechanisms drive sessile hematopoietic pocket formation and FBAH adhesion”) that the knockdown of Mp by RNAi produces a phenotype strikingly resembling that seen in the eater1 mutant (and thus that Mp is required for all attachment to endogenous hematopoetic environments) doesn't seem supported by the comparison of Figure panels 4A' to 4H'. While the dorsal patches seem to be gone in the Mp knockdown, it appears as if there is still a repeated hemocyte pattern on the lateral sides in the larvae shown. Having a close up of the lateral and dorsal niches for these genotypes instead of the GFP channel (whose relevance as a marker of the driver pattern is also not explained in the figure legends) would clarify this issue. It is clear there is a strongly reduced attachment in the absence of Mp, but if this the same in all the different endogenous hematopoetic locations is important to determine. Given that Mp has been reported to be expressed in the heart tube, next to where the dorsal patches are, and not to my knowledge near the PNS, which flanks the lateral ones, would fit with there being a variable effect in the different regions. This Figure has no quantitation, which also weakens this conclusion. If they do not have the data in hand to make the new figure/ do the quantitation, then they should soften the conclusion that this is the same as the eater1 phenotype and address textually that the effect on the lateral patches has not been rigorously assessed.

The experiments showing the disruption of the sessile compartment are now complemented with additional images, showing closeups of the dorsal and lateral patches (Figure 4N-O). Additionally, to provide better representation for the sessile tissue pattern, we now include quantifications for the relevant genotypes, and compare the stereotypical sessile tissue pattern with larvae where the compartment is disrupted (Figure 4K-M). We would like to note that on the overview images, it is possible to interpret the hemocytes as “repeated hemocyte pattern”, but in reality, this is due to the immobilization (cooling the larvae) and the settling of the circulating hemocytes. Since the contractions of the dorsal vessel diminish in low temperature, the cells in the circulation can accumulate between the muscle and epidermal layers, creating the impression of a repeated pattern (Figure 4N-O).

Reviewer #3:I believe this is a great paper, with exciting science, but I find it unnecessarily difficult to read and to assess critically. I was easily distracted while reading it and I may have missed important points.Essential revisions:1) Conceptually, it would be helpful to give the reasons for choosing to overexpress Atf3 in the fat body. I would also like to see more background information about this transcription factor in the Introduction.

See below.

2) I also miss a discussion about the possible physiological role of the system described here. Is Atf3 also involved in defining hemocyte docking sites elsewhere? Or should I regard Atf3 overexpression as an artificial but useful experimental system for the study of hematopoiesis?

The use of Atf3 was based on our previous observation of the hemocyte attachment to the fat body. The overexpression of Atf3 in the fat body at high levels is indeed artificial, and our goal was to exploit this as a platform to understand the fundamental mechanisms underlying hemocyte attachment to *Drosophila* tissues in an accessible manner. As the reviewer suggested, we now include additional background about Atf3 in the leading paragraph of the Result section.

3) In general, the figure legends give little or no information about the experiments shown. Instead they merely describe what conclusions I am expected draw, something that is anyway already explained in the text. There are many examples:a) In Figure 1, the reader has to guess how the different colors are generated. I assume that the green color comes from a UAS-GFP construct, driven by the C7 fat body-specific driver. An Hml:DsRed construct is apparently also involved. It should give red fluorescence to plasmatocytes. Apparently, that this has been converted to the "magenta" color, here labeled "Hemocyte" (I comment the "amber"-colored hemocytes below). Similar difficulties complicate interpretations in other figures too.b) The units on y-axes are sometimes unclear. Does "Number of FBAHs" (Figure 1, Figure 2, and Figure 3) refer to the number per unit area as defined in Figure 1—figure supplement 1? And what is "Circulating hemocytes (F.C.)" in Figure 1, Figure 1—figure supplement 1 and Figure 5?c) If "Hemese" (in Figure 2 and Figure 5) refers to antibody staining, for general visualization of hemocytes, then say so.d) What is "myr-RFP" (Figure 3A and B), and what does it signify?I will stop giving examples here, but my progress through the manuscript was slowed down dramatically by the fact that I constantly had to make guesses about what I could see in the figures.

We expanded the Figure legends and the Materials and methods section considerably to facilitate reading, including the specific suggestions of the reviewer.

4) How was the "amber"-coloring of hemocytes generated in Figure 1B (and elsewhere)? The text in the Materials and methods section seems to indicate that these cells were simply enhanced in this way via Photoshopping. That does give a striking effect, but is not valid evidence that these cells are attached to the fat body.

We added the description to the Materials and methods section. Briefly, larvae were fixed with the injection of 4% paraformaldehyde, immobilized on microscope slides and image with confocal microscope. Each individual confocal section was examined for GFP expression (fat body) and Hml:DsRed expression (hemocyte). The DsRed channel was locally recolored on each section when it was determined to overlap with the GFP signal. The attachment of the hemocytes is also demonstrated throughout the study on dissected fat bodies (e.g. Figure1—figure supplement 1).

5) On a similar note, the intracellular vacuoles in Figure 3F are colored "cyan", presumably by Photoshop, to generate the impression that they are contiguous with the similarly colored "pericellular" space under the basement membrane. This is a potentially dangerous way to fool the eye. Do the authors have evidence that these systems are contiguous, or that they are related in other ways? Is there evidence that the associated electron-dense material consists of extracellular matrix proteins?

We based our assessment on two previous publications, which used similar indication for the pericellular spaces, and concluded that the trapped material consists of ECM proteins. The representative Figure from Zang et al., 2015, shows the same type of distinction between pericellular spaces and cytoplasm/organelles, and points out the electron dense material as ECM protein aggregates.

6) As shown here, several extracellular matrix proteins are enriched over Atf3-expressing cells. Does that mean that the basement membrane is thickened? Zooming in on Figure 3F I get an impression that this may be the case.

We include a closeup image Author response image 1 to compare the two genotypes. According to our measurements in both genotypes the average thickness of the BM is 65-85 nms, which also corresponds to data published in Dai et al., 2017 (Figure S1B). What can create the impression of thicker BM is the proximity of the cytoplasm in *C7>Atf3WT* fat bodies. In controls the pericellular space between the cell membrane and the BM is considerably wider.

**Author response image 1. sa2fig1:** Transmission electron micrographs of control (left) and Atf3 overexpressing (right) adipocytes. Transgene and GFP expression was driven by the fat bodyspecific *C7-GAL4* driver. The average thickness of the BM is 65-85 nms in both genotypes. Basement membranes are indicated with arrowheads Scale bars: 2 μm.

7) I find no information about how the Atf3 clones were generated for Figure 1F, and very rudimentary information about the clones in Figure 3 and Figure 5.

We generated the clones using the hsFLPout system. We now include a description in the Materials and methods section and add description into respective Figure legends.

8) The magnification and resolution is not sufficient to show the filopodia and lamellipodia in Figure 2A, 2B, Figure 5G and 5H, even when I print out these figures over entire pages. They are even difficult to see when I zoom them in on the computer.

We now include higher magnification images of these samples with the filopodia and lamellipodia indicated.

9) Has the C7 driver ever been described? The authors refer to a previous article from the same lab (Rynes et al., 2012), where I only find a reference back to a paper by Grönke et al., (2003). However, C7 is not mentioned there. How specific is the C7 driver, and why is it used?

The *C7-GAL4* is a strong fat body-specific driver expressed in the fat body cells from early larval stages. A detailed description of the driver can be found in Koyama and Mirth, 2016, which we now refer to in the manuscript.

[Editors' note: further revisions were suggested prior to acceptance, as described below.]

Reviewer #4:This manuscript provides novel insight into the interaction of *Drosophila* blood cells, the hemocytes, with other tissues during the formation of sessile hemocytes clusters. The manuscript provides evidence on a key role of Multiplexin in the formation of these clusters, which, as this manuscript and previous work shows, are important sites for hemocyte proliferation and probably also differentiation. The reviewers in the first round of assessment have pointed out valid concerns regarding the manuscript and, in my opinion, the authors have done a good job addressing these concerns. I have only few remaining points/questions for the authors.On some occasions, the number of animals tested seem quite low, especially when *Drosophila* material is not exactly scarce. However, I don't see this as a decisive problem in this study, one reason being that differences between controls and experimental crosses are in general quite clear. Also, multiple experiments conducted back each other up. However, could authors provide more detailed information on how experiments were replicated? Specifically, were the animals studied from the progeny of one cross/the same patch of crosses or were experiments replicated temporally? This could be clarified in the Materials and methods section since it helps in assessing the reproducibility of the results.

All experiments represent at least two independent temporal replicates. We now include the sentence in Materials and methods section.